# A modular platform to display multiple hemagglutinin subtypes on a single immunogen

Dana Thornlow Lamson[1,2], Faez Amokrane Nait Mohamed[1], Mya Vu[1], Daniel P Maurer[1,2], Larance Ronsard[1], Daniel Lingwood[1], Aaron G Schmidt[1,2]*

[1]Ragon Institute of Mass General, MIT, and Harvard, Cambridge, United States; [2]Department of Microbiology, Harvard Medical School, Boston, United States

## eLife Assessment

This **valuable** manuscript describes the immunogenicity of a bead-on-a-string immunogen that allows the inclusion of multiple HA subtypes. The evidence to support the claims is **convincing**, and more importantly, this approach could be adapted to other vaccine platforms.

*For correspondence:
aschmidt@crystal.harvard.edu

**Abstract** Next-generation influenza vaccines aim to elicit cross-reactive humoral responses to multiple influenza subtypes. Such increased breadth would not only improve seasonal vaccines but may afford 'universal' protection against influenza subtypes, including those with pandemic potential. Here, we describe a 'beads-on-a-string (BOAS)' immunogen that tandemly links up to eight distinct hemagglutinin (HA) head domains from circulating and non-circulating influenzas. These BOAS are immunogenic in the murine model and elicit comparable serum responses to each individual component. Notably, we also find that BOAS elicit cross-reactive responses to influenza subtypes not included in the immunizing immunogen. Furthermore, BOAS conjugation to protein-based ferritin nanoparticles does not significantly augment serum responses suggesting that our BOAS platform is sufficient for eliciting cross-reactive responses without off-target effects induced by the nanoparticle scaffold. Finally, vaccination with a mixture of the same HA head domains is not sufficient to elicit the same neutralization profile as the BOAS immunogens or nanoparticles. This mix-and-match immunogen design strategy is a robust platform for eliciting responses to multiple influenza subtypes via a single immunogen, and a potential platform for other viral glycoproteins.

## Introduction

Influenza viruses, despite continuous surveillance and annual vaccination, are a significant public health problem resulting in ~500,000 deaths per year (*Wei et al., 2020*). This is largely a consequence of viral evolution and antigenic drift as it circulates seasonally within humans and ultimately impacts vaccine effectiveness. Additionally, the chance for spillover events from animal reservoirs (e.g. avian, swine) is increasing as population and connectivity also increase. To combat this, global surveillance is necessary to inform re-formulation of influenza vaccines to match circulating strains. However, long lead times for vaccine preparation in eggs and mutations acquired during that process (*Raymond et al., 2016*) can lead to vaccine mismatch and subsequent ineffectiveness. The SARS-CoV-2 pandemic has additionally underscored the need for pandemic preparedness as humans increasingly encounter animal reservoirs. Indeed, several avian influenzas, such as H5N1, H7N9, and H9N2, are currently being monitored for their pandemic potential. Thus, there is an urgent need to develop so-called

'universal' influenza vaccines that provide protection against both currently circulating (e.g. H1N1s, H3N2s, and Bs) and potentially pandemic viruses.

Several approaches have been taken to improve upon current influenza vaccines, largely focused on the influenza surface glycoprotein, hemagglutinin (HA). A major goal of immunogen design efforts is to elicit antibodies toward conserved epitopes on HA, including the receptor binding site (RBS) (*Ekiert et al., 2012*; *Hong et al., 2013*; *Krause et al., 2012*; *Krause et al., 2011*; *Lee et al., 2014*; *Lee et al., 2012*; *Schmidt et al., 2013*), trimer interface (TI) (*McCarthy et al., 2021*; *Bangaru et al., 2019*; *Watanabe et al., 2019*; *Dong et al., 2020*; *Bajic et al., 2019*), and stem (*Corti et al., 2011*; *Dreyfus et al., 2013*; *Dreyfus et al., 2012*; *Ekiert et al., 2009*; *Friesen et al., 2014*; *Kallewaard et al., 2016*); antibodies targeting such epitopes are often broadly protective. Indeed, RBS-directed antibodies often are potently neutralizing, as they directly inhibit HA engagement with host cell sialic acid, while TI- and stem-directed antibodies generally protect via Fc-dependent mechanisms (e.g. ADCC, ADCP). Several rational immunogen design approaches (*Caradonna and Schmidt, 2021*) such as epitope removal (*Impagliazzo et al., 2015*; *Yassine et al., 2015*), domain chimeras (*Nachbagauer et al., 2021*), computational sequence optimization (COBRAs) (*Giles and Ross, 2011*; *Skarlupka et al., 2020*; *Reneer et al., 2020*; *Wong et al., 2017*; *Carter et al., 2016*), epitope resurfacing (*Bajic et al., 2020*), and hyperglycosylation (*Bajic et al., 2019*; *Thornlow et al., 2021*; *Dosey et al., 2023*) attempt to focus on these epitopes.

An additional strategy to enrich for cross-reactive antibodies includes increasing antigen valency by displaying multiple antigens on a scaffold, such as a nanoparticle (*Bachmann and Jennings, 2010*; *Nguyen and Tolia, 2021*; *Curley and Putnam, 2022*). Such multivalency is thought to enhance immunogenicity and antigenicity by spacing antigens near one another to effectively cross-link B cell receptors (BCRs) (*Bachmann and Jennings, 2010*). Ferritin, which can display a total of 24 antigens at its eight threefold axes, has been used for influenza, Epstein-Barr virus, coronavirus, and HIV glycoproteins (*Joyce et al., 2022*; *Kanekiyo et al., 2013*; *Georgiev et al., 2018*; *He et al., 2016*; *Kanekiyo et al., 2019*; *McLeod et al., 2022*; *Kanekiyo et al., 2015*). For influenza, following immunization with a mixture of homotypic full-length seasonal H1, H3, and influenza B HA trimers displayed on ferritin nanoparticles, mice elicited responses with enhanced potency and breadth relative to the trivalent inactivated influenza vaccine (TIV) standard (*Kanekiyo et al., 2013*). Based on this success, they then designed 'mosaic' ferritin nanoparticles that displayed up to eight antigenically distinct H1 subtype HA receptor binding domains (RBDs) on a single particle, which outperformed admixtures of homotypic nanoparticles and elicited cross-reactive H1 broadly neutralizing antibodies (*Kanekiyo et al., 2019*). Further iterations of nanoparticle immunogens have co-displayed multiple full-length HA antigens on virus-like particles (VLPs) (*Cohen et al., 2021b*) and de novo designed particles (*Boyoglu-Barnum et al., 2021*), and the latter is currently in clinical trials (*Boyoglu-Barnum et al., 2021*, *Zhao et al., 2014*; *Etemad et al., 2008*).

An alternative approach for multivalent display involves tandemly linking antigens into a single polypeptide. This method may alleviate manufacturing challenges related to preparing multiple individual components for mosaic nanoparticles while still increasing size and valency of antigens. For example, using domain III from the dengue envelope surface glycoprotein, a bivalent (*Zhao et al., 2014*) or quadrivalent (*Etemad et al., 2008*) tandemly linked immunogen elicited neutralizing antibody responses to all four dengue serotypes and provided protection upon challenge. An analogous approach tandemly linked four different coronavirus RBDs into a single immunogen (*Hills et al., 2024*). These 'quartets' elicited responses to each individual components similar to a 'mosaic-4' nanoparticle that displayed monomeric RBDs on the surface of a nanoparticle (*Cohen et al., 2021a*; *Cohen et al., 2022*); these approaches were then combined and quartets displaying four or eight unique RBDs were multivalently displayed on nanoparticles. This combinatorial approach augmented both neutralization titers and cross-reactive responses.

Here, we tandemly linked multiple HA subtypes into a single immunogen via short, flexible linkers in a 'plug and play' platform we call 'beads-on-a-string' or BOAS. These recombinantly produced, protein-based immunogens have between three and eight unique HA heads, including circulating and non-circulating A and B influenzas. Mice immunized with either 3, 4, 5, 6, 7, or 8mer BOAS elicited responses to matched and mismatched HA components and could neutralize subsequent replication-restricted reporter (R3) viruses (*Creanga et al., 2021*). To further enhance valency, we generated two 4mer BOAS that included 8 different HA heads, conjugated both to ferritin nanoparticles using

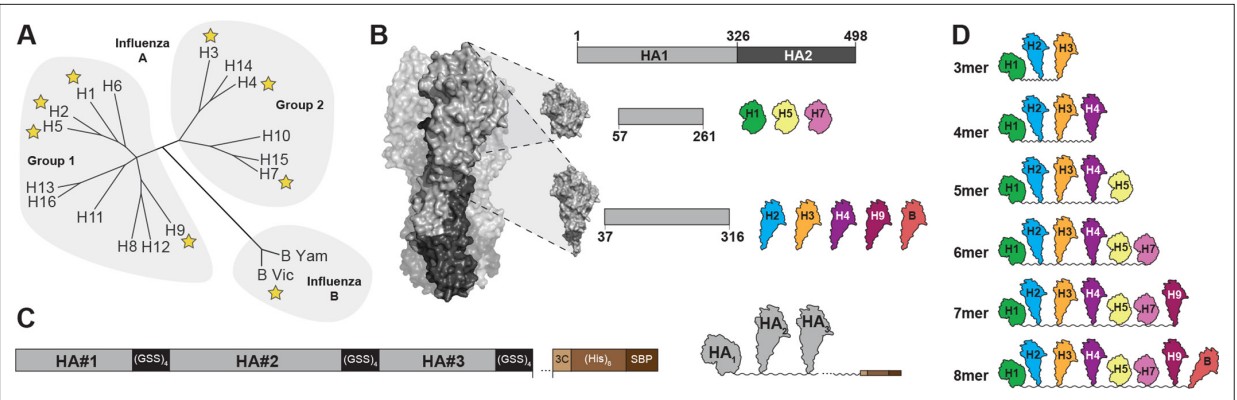

**Figure 1.** Design of beads-on-a-string (BOAS) immunogens. (**A**) Phylogenetic tree of influenza hemagglutinin (HA) subtypes. Subtypes are further categorized as group 1, group 2, or influenza B, and stars indicate subtypes included in the BOAS. (**B**) Native trimeric HA structure (A/H3/Hong Kong/1968; PDB: 4FNK) is shown on the left, with the HA1 domain in light gray, and HA2 in dark gray. A construct diagram of full-length soluble HA is shown at the top with the locations of head domain boundaries within HA1. Head domains of H1, H3, and H7 span residues 57–261, and head domains of H2, H3, H4, H9, and B span 37–316 of the HA1 region of HA. (**C**) Construct diagram of HA BOAS. BOAS are assembled by linking HA head domains (light gray) into a single polypeptide chain where HA head domains are separated by a flexible linker with four glycine-serine-serine (GSS) repeats. The C-terminus consists of an HRV-3C protease cleavable His-tag and streptavidin-binding-protein (SBP) tag for purification and other affinity tags. (**D**) BOAS used in this study.

The online version of this article includes the following source data and figure supplement(s) for figure 1:

**Figure supplement 1.** Adjusting flexible linker lengths on beads-on-a-string (BOAS).

**Figure supplement 1—source data 1.** Uncropped and labeled gel for *Figure 1B*.

**Figure supplement 1—source data 2.** Raw unedited gel for *Figure 1B*.

**Figure supplement 2.** Swapping 3mer order on beads-on-a-string (BOAS).

**Figure supplement 2—source data 1.** Uncropped and labeled gel for Figure Supplement 2C.

**Figure supplement 2—source data 2.** Raw unedited gel for Figure supplement 2C.

SpyTag-SpyCatcher (*Zakeri et al., 2012*; *Sutherland et al., 2019*), and showed that the serum responses elicited similar responses to the 8mer. Additionally, we showed that a mixture of the same HA head components was not sufficient to recapitulate the neutralizing responses elicited by the BOAS or BOAS NP. Collectively, these BOAS immunogens could elicit responses to multiple HA subtypes as a single immunogen and represents a general approach for multivalent display of other viral antigens.

## Results

### Design and characterization of beads-on-a-string (BOAS)

Monomeric HA heads representing antigenically distinct subtypes were tandemly linked together as a single linear construct, separated by a short, flexible glycine-serine-serine (GSS) linker, which we termed 'beads-on-a-string' or BOAS immunogens. Eight unique HA subtypes spanning group 1 and 2 influenza A viruses, as well as an influenza B virus, from seasonal, circulating, and non-circulating influenzas were selected to understand the permissibility of the platform to express different HAs (*Figure 1A*). We found that monomeric HA head domains varied in expression levels depending on the boundaries selected: H1, H5, and H7 HA monomers spanned residues 57–261 (H3 numbering), whereas H2, H3, H4, H9, and B HA monomers spanned residues 37–316 (H3 numbering) (*Figure 1B*). We then asked whether we could tandemly link multiple HA heads together as a single construct. Using linkers with either two, three, or four GSS repeats, we tandemly linked representative H1, H3, and B HAs (*Figure 1—figure supplement 1A*) together for recombinant expression in mammalian cells. Each construct, regardless of linker length expressed to similar levels (*Figure 1—figure supplement 1B*) and each individual HA component could be detected with a conformation-specific and subtype-specific monoclonal antibody (mAb) (*Figure 1—figure supplement 1C*). We next assessed whether the order of the HA heads adversely affected expression or accessibility to the conformation-specific

mAb. Using H5, H7, and H9 HA heads, we created two constructs: (1) H5-H9-H7 and (2) H9-H5-H7 each with a four GSS linker. Each construct had comparable levels of expression and reactivity to the respective mAbs (*Figure 1—figure supplement 2*).

Based on these initial data, we next designed six constructs with three to eight individual HA head components from representative H1, H2, H3, H4, H5, H7, H9, and B influenzas; to maximize spacing between each HA component, we used the 4 x GSS linker. Each BOAS was cloned into a single poly-peptide chain with a protease-cleavable C-terminal purification tag (*Figure 1C*). We synthesized six different BOAS, and define each construct as 3mer, 4mer, etc., referencing the number of HA components (*Figure 1D*). Each BOAS construct could be purified to homogeneity as assayed by SDS-PAGE after immobilized metal affinity chromatography (*Figure 2A*) and correspond to the expected molecular weight following proteolytic cleavage of native glycans with PNGase-F (*Figure 2B*). After size-exclusion chromatography (SEC), the non-PNGase-treated BOAS were monodisperse (*Figure 2C*). Following SEC, affinity tags were removed with HRV-3C protease; cleaved tags, uncleaved BOAS, and His-tagged enzyme were removed using cobalt affinity resin and snap frozen in liquid nitrogen before immunizations. BOAS maintained monodispersity upon thawing, though over time, degradation was observed following longer term (>1 week) storage at 4 °C (*Figure 2—figure supplement 1*). This degradation became more significant as BOAS increased in length (*Figure 2—figure supplement 1*).

We then analyzed the BOAS using negative stain electron microscopy (NS-EM). Individual HAs were readily discernible and equal to the number of components in the respective BOAS (*Figure 2D and E*); we observed that the 3mer could adopt both extended and collapsed, triangular-like conformations (*Figure 2D*), whereas longer BOAS, such as the 8mer, adopted a rosette-like conformation (*Figure 2E*). This is likely a consequence of the flexible GSS linker separating the individual HA head components, as well as the addition of significantly more HA head components to the construct.

To assess conformational integrity and ensure that each HA present was properly folded, we assembled a panel of conformation- and subtype-specific mAbs to assay each individual component. This included mAb S5V2-29 (*Watanabe et al., 2019*), a broadly-reactive TI-directed positive control antibody and mAbs 5J8 (*Hong et al., 2013*; *Krause et al., 2011*), 2G1 (*Krause et al., 2012*), K03.12 (*McCarthy et al., 2018*), P2-D9 (*Caradonna et al., 2022*; *Figure 2—figure supplement 2*), H5.3 (*Winarski et al., 2015*), H7.167 (*Thornburg et al., 2016*), and H1209 (*Bajic and Harrison, 2021*) which are specific to H1, H2, H3, H5, H5, H7, and B subtypes, respectively; an H9 conformation-specific mAb was not available. Each individual HA component was detected using subtype-specific mAb in ELISA, and overall affinity for each component was comparable regardless of BOAS length (*Figure 2F*). Collectively, these data support a plug-and-play platform that can readily exchange HA components while retaining conformational integrity and accesibility.

## Immunogenicity of BOAS

To determine immunogenicity of each BOAS immunogen, we performed a prime-boost-boost vaccination regimen in C5BL/6 mice at two-week intervals with 20 µg of immunogen and adjuvanted with Sigma Adjuvant (*Figure 3A*). We compared these BOAS to a control group immunized with a mixture of the eight HA heads included in the 8mer. We then tested serum reactivity to the BOAS immunogen as well as individual full-length soluble HA trimers of each component. Serum titers against the immunogen were detectable after a single prime, and increased following a boost, and stayed relatively consistent until d42 (*Figure 3B*). Serum responses varied for differing BOAS lengths, with the 4mer and 5mer BOAS being the most immunogenic (*Figure 3C*). Each of the BOAS, regardless of its length, elicited binding titers to all matched full-length HAs representing individual components (*Figure 3D*). For example, the 6mer, which contains heads of H1, H2, H3, H4, H5, and H7, elicited reactivity to matched HAs. Interestingly, some BOAS elicited cross-reactivity to mismatched components not present in the immunogen. For example, the 3mer, which contained H1, H2, and H3 HA heads, elicited detectable titers to H4 and H5 HAs (*Figure 3D*). Additionally, all groups elicited binding titers to mismatched historical H3 and H1 HAs (*Figure 3D*). When comparing the 8mer and mix groups, the 8mer tended to elicit elevated binding titers to matched and mismatched components than an equi-molar mixture of the same components (*Figure 3E*). Similar binding trends were also observed with d28 serum, though the difference between the 8mer and mix groups was more pronounced at d28 (*Figure 3—figure supplement 1*).

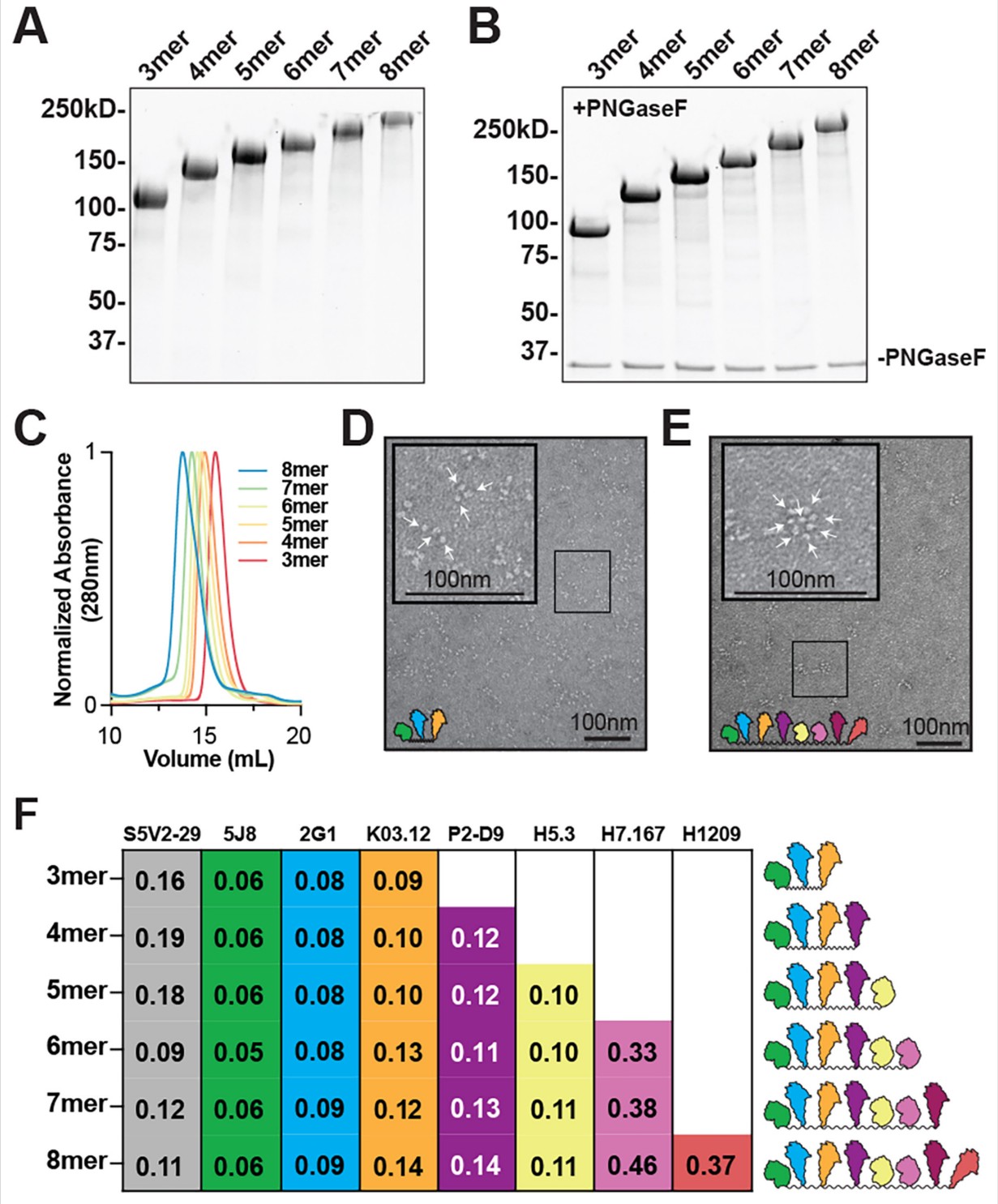

**Figure 2.** Biochemical characterization of beads-on-a-string (BOAS). (**A**) SDS-PAGE analysis of 3mer-8mer BOAS under non-reducing conditions. (**B**) SDS-PAGE analysis of 3mer-8mer BOAS under non-reducing conditions following glycan digestion with PNGase-F. (**C**) Size exclusion chromatography traces of 3mer-8mer BOAS on a Superdex 200 column in PBS. (**D**) Negative stain electron microscopy (NS-EM) images of the 3mer. Arrows point to individual HA monomers. Inset image is 2.5 X zoomed, enlarged image of boxed area. (**E**) NS-EM images of the 8mer. Arrows point to individual HA monomers. Inset image is 2.5 X zoomed, enlarged image of boxed area. (**F**) Characterization via ELISA of BOAS components with subtype-specific antibodies. Apparent $K_D$ values in μM are reported for each antibody for each BOAS immunogen. Antibodies corresponding to each component are as follows: S5V2-29 (*Watanabe et al., 2019*) (interface, all components except B/Mal/04), 5J8 (*Hong et al., 2013*; *Krause et al., 2011*) (RBS, H1), 2G1

*Figure 2 continued on next page*

*Figure 2 continued*

(*Krause et al., 2012*) (RBS, H2), K03.12 (*McCarthy et al., 2018*) (RBS, H3), P2-D955 (H4), H5.3 (*Winarski et al., 2015*) (RBS, H5), H7.167 (*Thornburg et al., 2016*) (RBS periphery, H7), H1209 (*Bajic et al., 2020*) (RBS, **B**).

The online version of this article includes the following source data and figure supplement(s) for figure 2:

**Source data 1.** Uncropped and labeled gel for *Figure 2A and B*.

**Source data 2.** Raw unedited gel for *Figure 2A and B*.

**Source data 3.** Uncropped images for *Figure 2D*.

**Source data 4.** Uncropped images for *Figure 2E*.

**Source data 5.** Raw ELISA data for *Figure 2F*.

**Figure supplement 1.** Size exclusion chromatography traces of purified beads-on-a-string (BOAS) immunogens over time.

**Figure supplement 2.** P2-D9 mAb specificity for H4.

## Cross-reactivity of immune responses

To understand potential cross-reactivity observed in the serum analyses, we used a Consurf-like (*Glaser et al., 2003*; *Ashkenazy et al., 2016*) method to assess sequence conservation across the eight HA components in the BOAS. Each component of the BOAS was aligned structurally to the H2 head domain and scored at each position for amino acid conservation at each position on a scale of 1–9. These were then pseudo-colored on the structure to visualize conserved and variable epitopes on the head domain surface (*Figure 4A*). We observed regions of both significant variability as well as conservation, notably the TI epitope, as well as the core sialic acid interacting residues in the RBS, the latter which remained nearly conserved in all BOAS (*Figure 4A*). This was quantified by taking the average amino acid conservation score across both the RBS and TI epitopes as well as the entire sequence, and we see that these epitopes retain higher conservation relative to the overall sequence as BOAS length increases. We then determined approximate degrees of focusing to the TI and RBS via a serum competition ELISA using mAbs. All the BOAS maintained, approximately, between 15–35% focusing to the TI epitope, whereas focusing on the RBS was more variable (*Figure 4C*). Both RBS focusing (*Figure 4D*) and TI focusing (*Figure 4E*) trended similarly with homology scores for each epitope, indicating epitope conservation may be influencing immune focusing to particular epitopes.

## Design of BOAS-conjugated nanoparticles

To further increase avidity effects, as well as potential immunogenicity, we split our 8mer BOAS into two 4mer BOAS and attached both in an equimolar ratio to *H. pylori* ferritin nanoparticles (NP) using SpyTag-SpyCatcher (*Zakeri et al., 2012*), herein referred to as BOAS-NPs (*Figure 5A*). The two 4mer BOAS efficiently conjugated to the NP as determined by SEC and SDS-PAGE (*Figure 5—figure supplement 1*). Further biophysical characterization using dynamic light scattering (DLS) showed a shift from 9.8±1.1 nm to 21.3±2.0 nm and NS-EM showed spherical particles with projections (*Figure 5D and E*). We verified the presence and structural integrity of each BOAS component on the NP using an ELISA with the same panel of conformation- and subtype-specific mAbs (*Figure 5E*).

## Immunogenicity of BOAS-conjugated nanoparticles

We next immunized mice with BOAS-NP and a SpyCatcher NP control to assess immunogenicity. Mice were immunized with an equal amount of NP relative to the BOAS (20 µg), in an equivalent homologous prime-boost-boost regimen (*Figure 6A*). The BOAS-NP was significantly more immunogenic than the SpyCatcher NP control, eliciting titers approximately an order of magnitude greater by d42 (*Figure 6B*). The BOAS NP, however, elicited significantly reduced titers to the SpyCatcher NP scaffold relative to the control (*Figure 6C*). When each group was evaluated for titers against both the BOAS components, the SpyCatcher nanoparticle scaffold, or BOAS-NP (*Figure 6D and E*), the BOAS-NP elicited equivalent titers to each set of BOAS, and greater titers to the decorated nanoparticle relative to the ferritin scaffold (*Figure 6E*). When we examined titers to individual HA components, we observed detectable titers over baseline to all eight HAs present on the BOAS-NPs (*Figure 6F*). When compared to the immunogenicity of the 3mer to 8mer BOAS immunogens, the BOAS-NP showed comparable titers to each component relative to BOAS that contained a given component (*Figure 6G*).

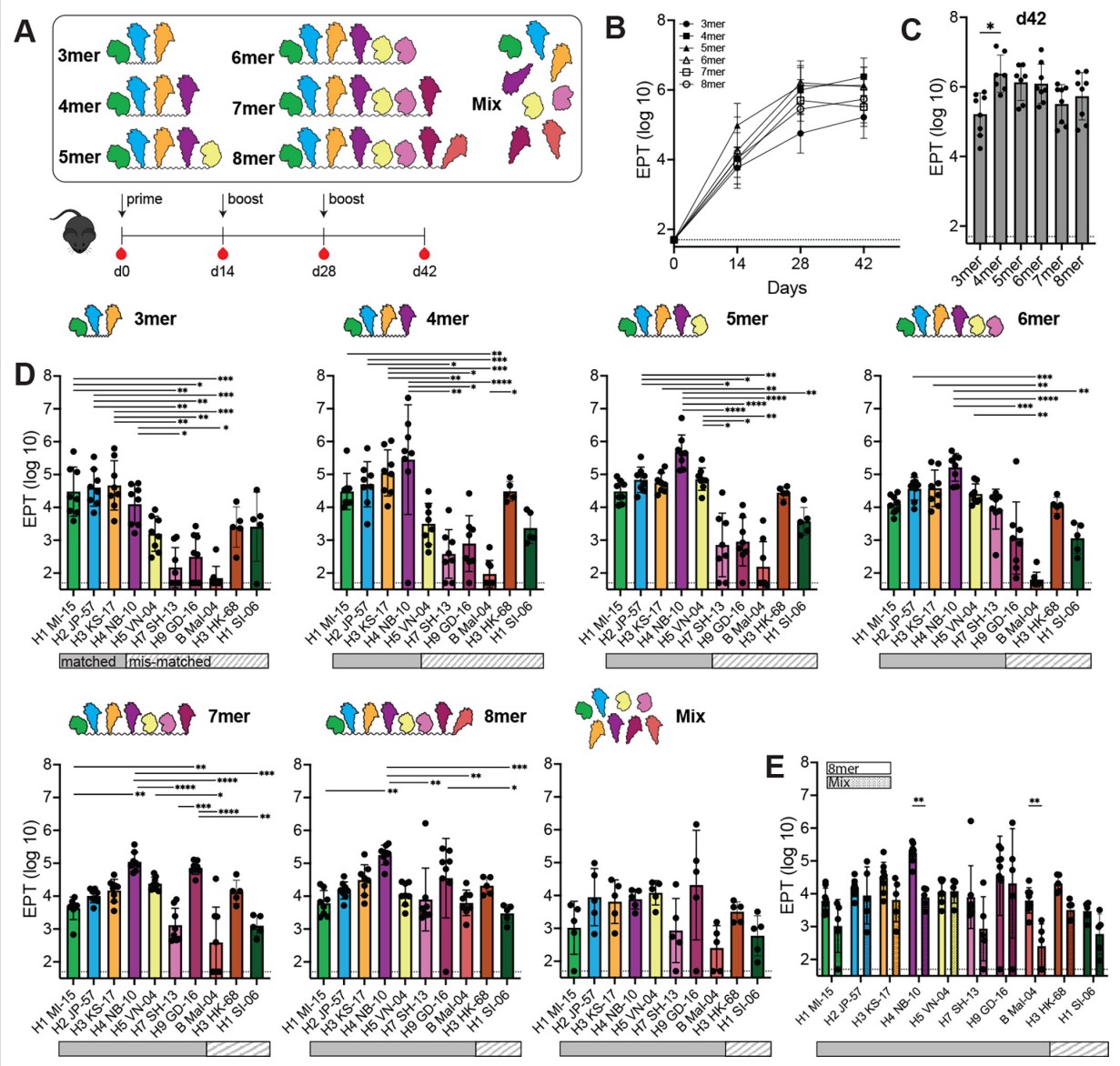

**Figure 3.** Murine immunizations with 3mer-8mer beads-on-a-string (BOAS). (**A**) Schematic of immunization regimen with 3mer-8mer BOAS and a Mix control containing an equal molar of the eight individual hemagglutinin (HA) heads not connected with a linker. Mice were primed with 20 µg of immunogen, followed by two homologous boosts two weeks apart. (**B**) Immunogenicity of BOAS over time as determined by serum ELISA. Serum reactivity to each immunogen is reported as the endpoint titer (EPT) dilution factor for each group (n=8) and d14, d28, and d42. Data points are an average of n=8 mice and error bars are +/-1 s.d. (**C**) Serum titers to each BOAS at the final time point, d42. Data points for single time point antigen titers are from individual mice n=8, bars represent mean titers and error bars are +/-1 s.d., *=p <0.05 as determined by Kruskal-Wallis test with Dunn's multiple comparison post hoc test relative to the 3mer. (**D**) Serum titers to full-length HA trimers elicited by 3mer-8mer BOAS. Solid bars below each plot indicate a matched subtype, and striped bars indicate a mismatched subtype (i.e. not present in the BOAS). (**E**) Side-by-side comparison of 8mer (solid bars) and Mix (dotted bars) serum titers to individual HA components. Data points are serum EPTs from individual mice (n=8) and error bars are +/-1 s.d., *=p<0.05, **=p<0.01, ***=p<0.001, ****=p<0.0001 as determined by a Kruskal-Wallis test with Dunn's multiple comparison post hoc test.

The online version of this article includes the following source data and figure supplement(s) for figure 3:

**Source data 1.** Raw ELISA for *Figure 3B–E*.

**Figure supplement 1.** Day 28 serum reactivity from beads-on-a-string (BOAS) and BOAS NP Cohorts.

**Figure supplement 1—source data 1.** Raw ELISA data for *Figure 3—figure supplement 1A and B*.

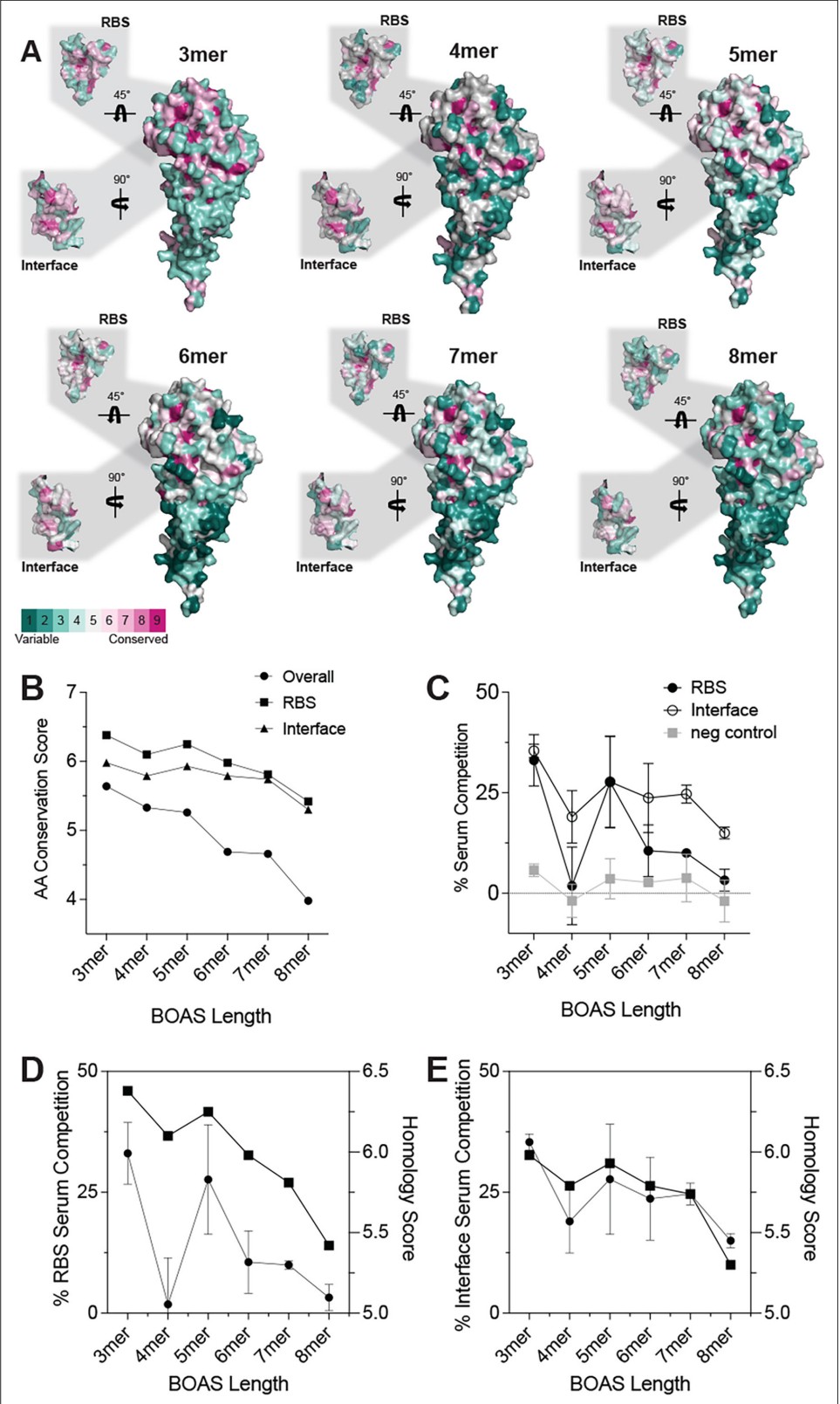

**Figure 4.** Beads-on-a-string (BOAS) component epitope homology and serum competition. (**A**) Amino acid conservation of each BOAS projected on the A/H2/Japan/1957 head domain structure (PDB: 2WRE). Variability and conservation scores were determined as the % conservation of each residue at each position for each BOAS composition. Cut-outs are RBS and Trimer interface (TI) epitopes with their respective conservation. (**B**)

*Figure 4 continued on next page*

*Figure 4 continued*

Homology score as determined by the average % conservation of each residue across the entire protein sequence (overall, circle), and the RBS (square) and TI (triangle) epitopes. (**C**) Serum competition to RBS and TI epitopes as determined via competition with a cocktail of RBS-directed antibodies, TI-directed antibodies, or a stem-directed negative control antibody. % competition is determined by the relative decrease in serum binding to each BOAS normalized to an untreated control. Each data point represents a mean from n=3 mice and error bars are +/-1 s.d. (**D**) Trend overlay of % serum competition with RBS-directed antibodies and homology score of the RBS epitope as a function of BOAS length. (**E**) Trend overlay of % serum competition with TI-directed antibodies and homology score of the TI epitope as a function of BOAS length.

The online version of this article includes the following figure supplement(s) for figure 4:

**Figure supplement 1.** Sequence identity matrices of beads-on-a-string (BOAS) components.

## Neutralization of influenza viruses by BOAS-elicited sera

We next determined serum neutralization titers to matched and mismatched influenza viruses based on the BOAS components (*Figure 7*). We tested serum from mice immunized with BOAS against H1/Michigan/2015 H1N1 and H3/Kansas/2017 H3N2 viruses, whose HA was present in all the BOAS. These showed varying neutralization titers to both viruses, with all matched BOAS neutralizing virus except for the 7mer and H1N1 virus. Additionally, we tested serum against H5/Vietnam/2004 H5N1 and H7/Shanghai/2013 H7N9 viruses. Neutralization of H5N1 virus was variable, though individual mice from 4mer, 5mer, 6mer, 7mer, 8mer, and BOAS NP groups were able to strongly neutralize this virus. Weak neutralization titers were detected for 6mer, 7mer, and 8mer BOAS. This neutralization was only slightly above the high background neutralization of the negative control NP serum. Serum from the BOAS-conjugated nanoparticle group elicited similar neutralization titers relative to the BOAS alone. In most cases, neutralization titers elicited from the BOAS and BOAS NP cohorts were greater than that from the mix cohort. The scaffold control serum did not neutralize any viruses tested.

## Discussion

Here, we engineered BOAS that included tandemly linked, antigenically distinct HA heads as a single construct. This platform allows a mixing-and-matching of up to eight distinct HA heads from both influenza A and B viruses. Furthermore, we showed that the order and number of HA heads can vary without losing reactivity to conformation-specific mAbs in vitro, highlighting the flexibility of this platform. Mice immunized with BOAS had comparable serum reactivity to each individual component though relative binding and neutralization titers varied between immunogens; this is likely a consequence of length and/or composition. Further oligomerization for increased multivalent display was accomplished by conjugating two 4mer BOAS inclusive of eight distinct HA heads to a ferritin nanoparticle via SpyTag/SpyCatcher ligation. Similar to the BOAS, these conjugated nanoparticles elicited similar titers to all eight HA components and could neutralize matched viruses.

Thus, tandemly linking HA heads is a robust method for displaying multiple influenza HA subtypes in a single protein-based immunogen. Binding titers were elicited to all components present in the immunogen, and there was no significant correlation between HA position within the BOAS (i.e. internal or terminal) and immunogenicity. However, the relative immunogenicity of each HA varied despite equimolar display of each HA subtype. There were qualitatively immunodominant HAs, notably H4 and H9, and these were relatively consistent across BOAS in which they were a component; this effect was reduced in the mix cohort. Further studies using the modularity of the BOAS could further deconvolute relative immunodominances of HA subtypes.

Despite similar binding titers across multiple BOAS lengths, expression levels and neutralization titers were quite variable. While all 3mer to 8mer BOAS could be overexpressed, expression inversely correlated with overall length. To mitigate this, multiple BOAS (e.g. two 4mers) or conjugation to protein-based nanoparticles, as was done here, could be used to ensure coverage of each desired HA subtype. Furthermore, neutralization titers were quite variable across different BOAS lengths despite similar binding titers. This may be related to multiple factors, including homology, stability, and accessibility of neutralizing epitopes for different BOAS lengths. Notably, for longer BOAS, we observed degradation following longer term storage at 4° C, which may reflect their overall stability. Studies

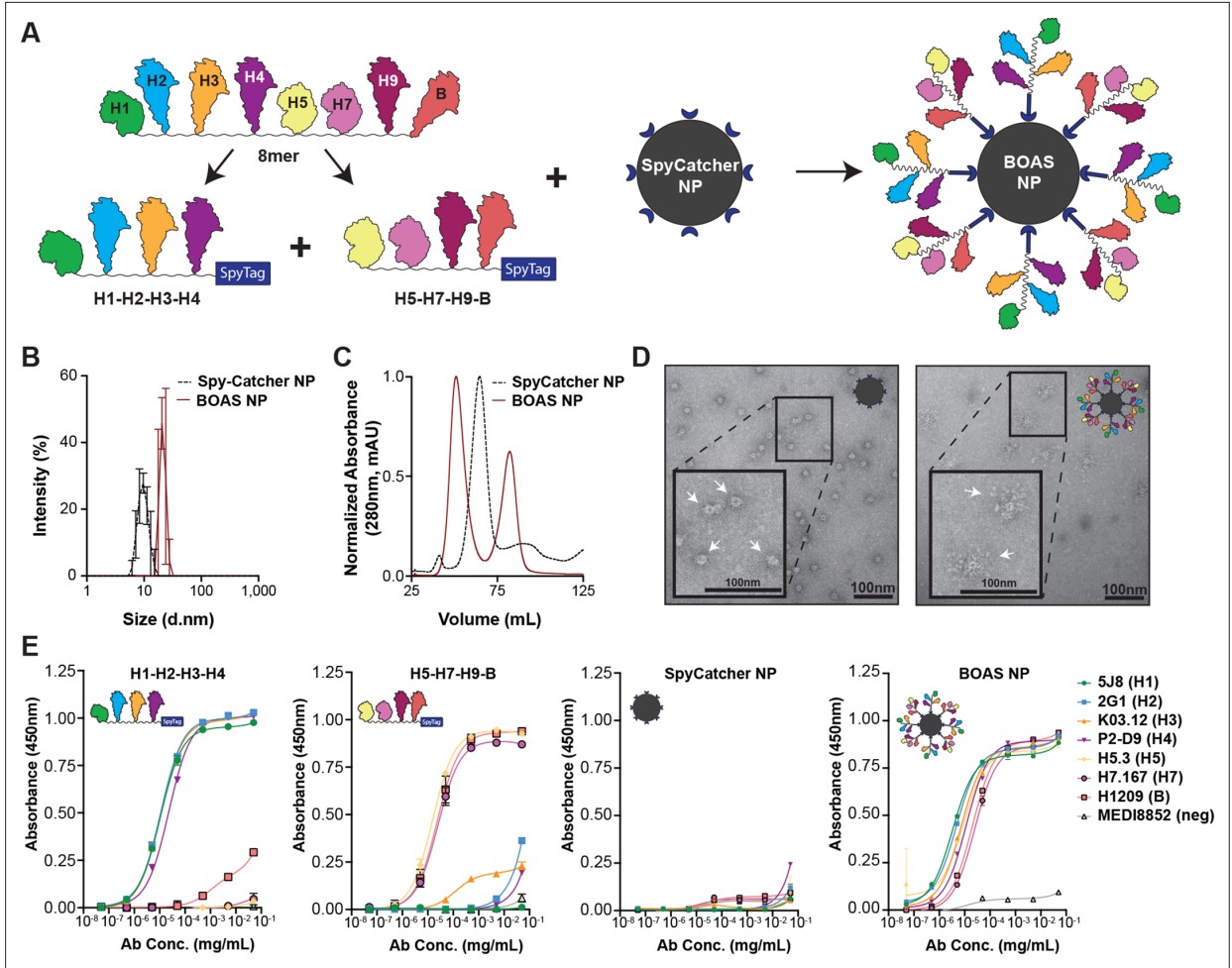

**Figure 5.** Design of beads-on-a-string (BOAS) nanoparticle (NP) immunogens. (**A**) Schematic of design of BOAS nanoparticles (NP). The eight original 8mer components were split into two 4mers, the first containing H1, H2, H3, and H4 (H1–H2–H3–H4), and the second containing H5, H7, H9, and B (H5–H7–H9–B). Each were expressed with an N-terminal SpyTag, then attached at equimolar ratios to an N-terminally fused SpyCatcher ferritin nanoparticle (SpyCatcher NP). This yields a NP with both 4mer BOAS conjugated to its surface (2x4 mer BOAS NP). (**B**) Dynamic light scattering (DLS) distributions of SpyCatcher NP and 2x4 mer BOAS NP. Curves represent an average of three technical replicate measurements and error bars are +/-1 s.d. (**C**) Size exclusion traces of SpyCatcher NP and 2x4 mer BOAS NP on an S400 column. (**D**) Representative negative stain electron microscopy (NS-EM) image of SpyCatcher NP. Arrows indicate individual nanoparticles, and the inset is a 2 X zoom of the boxed area. Representative NS-EM image of BOAS NP. Arrows indicate individual nanoparticles, and the inset is a 2 X zoom of the boxed area. (**E**) ELISA of BOAS components, SpyCatcher NP, and 2x4 mer BOAS NP with structure-dependent subtype-specific mAbs and a negative control stem-directed mAb, MEDI8852 (*Kallewaard et al., 2016*). Antibodies corresponding to each component are as follows: 5J8 (*Hong et al., 2013*; *Krause et al., 2011*) (RBS, H1), 2G1 (*Krause et al., 2012*) (RBS, H2), K03.12 (*McCarthy et al., 2018*) (RBS, H3), P2-D9 (*Caradonna et al., 2022*) (H4), H5.3 (*Winarski et al., 2015*) (RBS, H5), H7.167 (*Thornburg et al., 2016*) (RBS periphery, H7), H1209 (*Bajic and Harrison, 2021*) (RBS, B). ELISA data points are technical duplicates and error bars are +/-1 s.d.

The online version of this article includes the following source data and figure supplement(s) for figure 5:

**Source data 1.** Uncropped images of *Figure 5D*.

**Source data 2.** Raw ELISA data for *Figure 5E*.

**Figure supplement 1.** Nanoparticle conjugation efficiency SDS-PAGE.

**Figure supplement 1—source data 1.** Uncropped and labeled gel for *Figure 5—figure supplement 1*.

**Figure supplement 1—source data 2.** Raw unedited gel for *Figure 5—figure supplement 1*.

manipulating BOAS composition at intermediate lengths could optimize neutralizing responses to particular influenzas of interest.

Based on the immunogenicity of the various BOAS and their ability to elicit neutralizing responses, it may not be necessary to maximize the number of HA heads into a single immunogen. Indeed, it qualitatively appears that the intermediate 4-, 5-, and 6mer BOAS were the most immunogenic

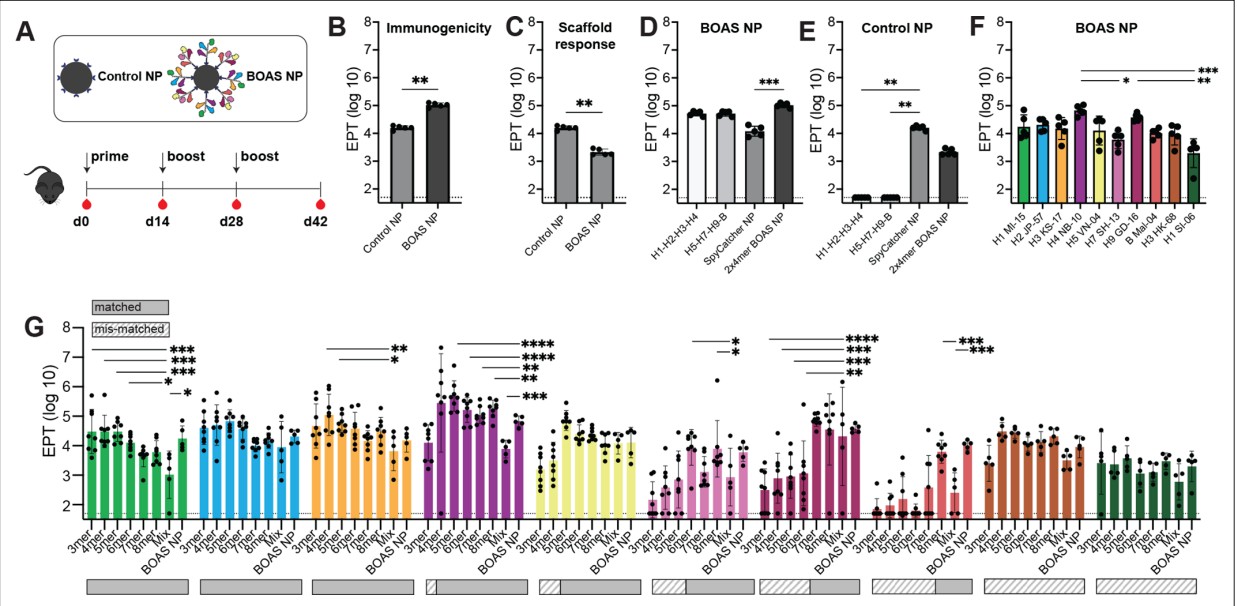

**Figure 6.** Murine immunizations with beads-on-a-string (BOAS) nanoparticles (NP). (**A**) Schematic of immunization regimen with 2x4 mer BOAS NP and SpyCatcher NP control. Mice were primed with 20 µg of NP, followed by two homologous boosts two weeks apart. (**B**) Serum reactivity to each immunizing antigen at d42. **p<0.01 as determined by the Mann-Whitney test. (**C**) Serum reactivity to Control NP scaffold by both groups. **p<0.01 as determined by the Mann-Whitney test. (**D**) Serum reactivity elicited by BOAS NP group (n=5) at d42 to both individual BOAS components (H1–H2–H3–H4 and H5–H7–H9–B), Control NP, and BOAS NP. *** p<0.001 as determined by the Kruskal-Wallis test with Dunn's multiple comparison post hoc test. (**E**) Serum reactivity elicited by Control NP group (n=5) at d42 to both individual BOAS components (H1–H2–H3–H4 and H5–H7–H9–B), Control NP, and BOAS NP. ** P<0.01 as determined by the Kruskal-Wallis test with Dunn's multiple comparison post hoc test. (**F**) Serum reactivity elicited by BOAS NP group (n=5) to individual full-length HA components of BOAS at d42. *=p<0.05, **=p < 0.01, ***=p < 0.001 as determined by the Kruskal-Wallis test with Dunn's multiple comparison post hoc test. (**G**) Serum reactivity to each matched and mis-matched individual full-length HA component of BOAS (H1/MI/15 light green, H2/JP/57 blue, H3/KS/17 light orange, H4/NB/10 purple, H5/VN/04 yellow, H7/SH/13 pink, H9/GD/16 maroon, B/ Mal/04 salmon, H3/HK/68 dark orange, and H1/SI/06 dark green) by 3mer-8mer BOAS, Mix control, and BOAS NP. Solid bars below each plot indicate a matched subtype, and striped bars indicate a mismatched subtype (i.e. not present in the BOAS, Mix, or NP). *=p<0.05, **=p<0.01, ***=p<0.001, ****=p<0.0001 as determined by the Kruskal-Wallis test with multiple comparisons relative to the mix control group. In all plots, data points for single time point antigen titers are from individual mice (n=5 for BOAS NP and Mix cohorts, n=8 for 3mer-8mer BOAS studies), bars represent mean titers and error bars are +/-1 s.d.

The online version of this article includes the following source data for figure 6:

**Source data 1.** Raw ELISA data for *Figure 6B–F*.

**Source data 2.** Raw ELISA data for *Figure 6G*.

and this length may be sufficient to effectively engage and crosslink B cell receptors (BCRs) for potent stimulation. These BOAS also had similar or improved binding cross-reactivity to mismatched HAs as compared to longer 7- or 8mer BOAS. Notably, the 3mer BOAS elicited detectable cross-reactive binding titers to H4 and H5 mismatched HAs. This observed cross-reactivity could be due to sequence conservation between the HAs, as H3 and H4 share ~51% sequence identity, and H1 and H2 share ~46% and~62% overall sequence identity with H5, respectively (***Figure 4—figure supplement 1***). Additionally, the degree of surface conservation decreased considerably beyond the 5mer as more antigenically distinct HAs were added to the BOAS. These data suggest that both antigenic distance between HA components and BOAS length play a key role in eliciting cross-reactive antibody responses, and further studies are necessary to optimize BOAS valency and antigenic distance for a desired humoral response.

Potential enrichment of serum antibodies targeting the conserved RBS and TI epitopes may also be contributing to observed cross-reactivity. Both epitopes are relatively conserved across all BOAS (***Figure 4C***), and the two BOAS showing the most cross-reactivity, the 3mer and 5mer, elicit a significant portion of the serum response toward both RBS and TI epitopes as determined via a serum competition assay with available epitope-directed mAbs (***Figure 4B***). Notably, this proportion is approximate, as at the time of reporting, mAbs that bind the receptor binding site

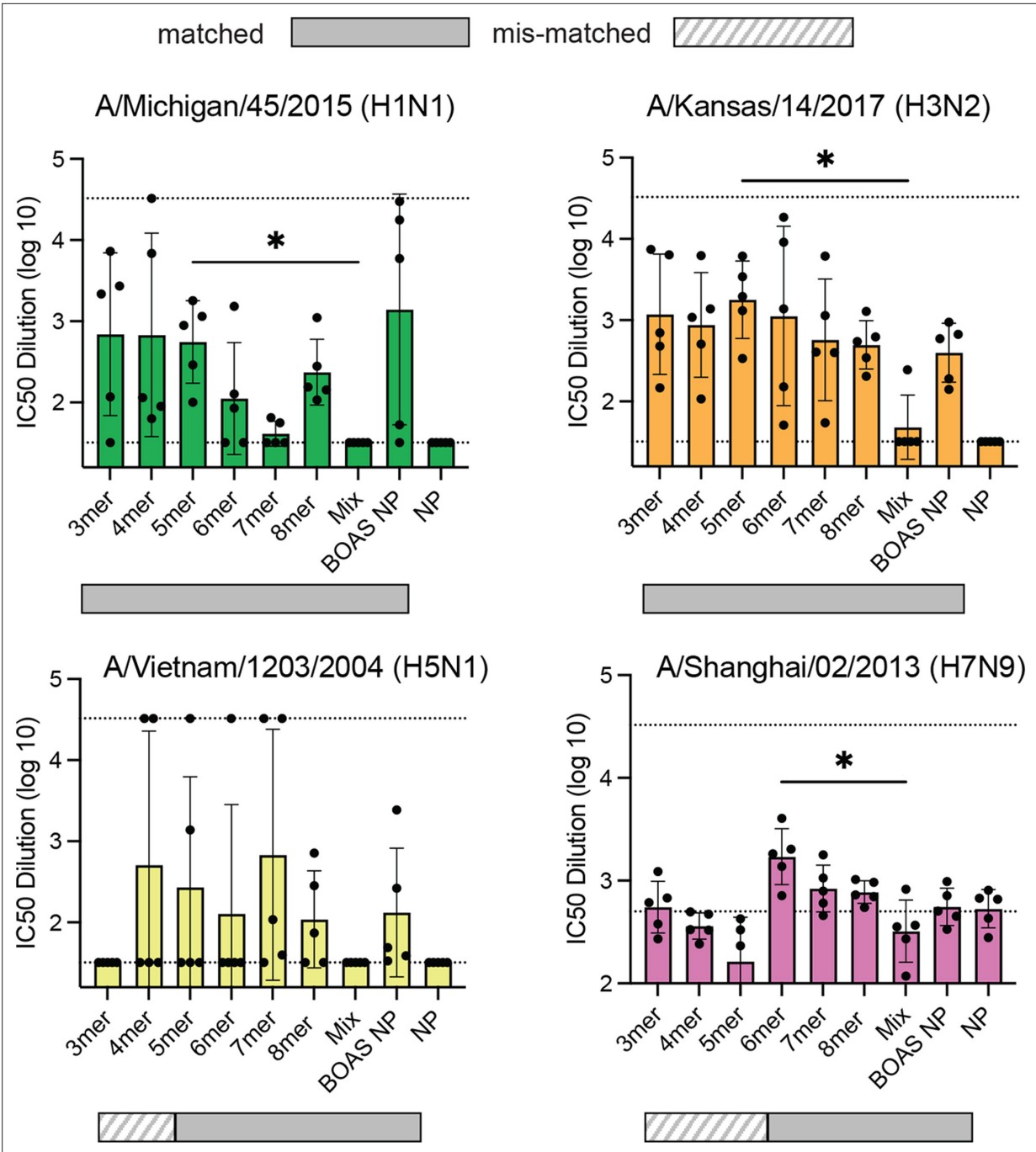

**Figure 7.** Microneutralization titers to matched and mismatched virus. Microneutralization of matched and mis-matched pseudo viruses: H1N1 (green, top left), H3N2 (orange, top right), H5N1 (yellow, bottom left), and H7N9 viruses (pink, bottom right) with d42 serum. Solid bars below each plot indicate a matched subtype, and striped bars indicate a mismatched subtype (i.e. not present in the BOAS). Nanoparticles (NP negative controls) were used to determine the threshold for neutralization. Upper and lower dashed lines represent the first dilution (1:32) (for H1N1, H3N2, and H5N1) or neutralization average with negative control NP serum (H7N9), and the last serum dilution (1:32,768), respectively, and points at the dashed lines indicate $IC_{50}$s at or outside the limit of detection. Individual points indicate $IC_{50}$ values from individual mice from each cohort (n=5). The mean is denoted by a bar and error bars are +/-1 s.d., *=$p<0.05$ as determined by a Kruskal-Wallis test with Dunn's multiple comparison post hoc test relative to the mix group.

The online version of this article includes the following source data for figure 7:

**Source data 1.** Raw microneutralization data for *Figure 7*.

of all components were not available. RBS-directed mAbs to the H4 and H9 components were not available, and the RBS-directed antibodies used targeting the other HA components have different footprints around the periphery of the RBS. Additionally, there are currently no reported influenza B TI-directed mAbs in the literature. Therefore, this may be an underestimate of the serum proportion focused on the conserved RBS and TI epitopes. Isolated TI-directed mAbs, in particular, can engage more than nine unique subtypes across both group 1 and 2 influenzas (*McCarthy et al., 2021*; *Watanabe et al., 2019*), and our monomeric head-based BOAS immunogens have the otherwise occluded TI epitope exposed (*McCarthy et al., 2021*; *Bangaru et al., 2019*; *Watanabe et al., 2019*). Furthermore, we have previously shown that this TI epitope, when exposed, is immunodominant in the murine model (*Bajic et al., 2019*). Further studies with different combinations of HAs could aid in understanding how length and composition influences epitope focusing. For example, a BOAS design with a cluster of group 1 HAs followed by a cluster of group 2 HAs, rather than our roughly alternating pattern could influence which HAs are in close proximity to one other or could be potentially shielded in certain conformations and thus could affect antigenicity. Combining the BOAS platform with other immune-focusing approaches (*Dosey et al., 2023*), such as hyperglycosylation (*Bajic et al., 2019*; *Thornlow et al., 2021*; *Ingale et al., 2014*; *Eggink et al., 2014*) or resurfacing (*Bajic et al., 2020*; *Hai et al., 2012*) could enhance cross-reactive responses. Additionally, modifying linker spacing and rigidity can also be used as a mechanism to enhance BCR cross-linking and thus enhance cross-reactive B cell activation and elicitation (*Veneziano et al., 2020*).

BOAS can be further multimerized via conjugation to a surface of a NP. Interestingly, this only had a marginal effect on immunogenicity. The BOAS NP elicited titers of ~$10^5$ (*Figure 6B*), whereas the best BOAS alone reached an order of magnitude greater (*Figure 3C*). This appears in contrast with other studies where attaching an antigen to a NP scaffold enhanced immunogenicity and neutralization potency (*Kanekiyo et al., 2013*; *Jardine et al., 2013*; *Tokatlian et al., 2019*; *Kato et al., 2020*; *Marcandalli et al., 2019*). One recent example designed quartets of antigenically distinct SARS-like betacoronavirus receptor binding domains (RBDs) coupled to an mi3-VLP scaffold via a similar SpyTag-SpyCatcher system and showed increased binding and neutralization titers following conjugation to the NP compared to quartet alone (*Hills et al., 2024*). This discrepancy may be in part due to the larger mi3 NP (*Bruun et al., 2018*) which displays 60 copies of the antigen rather than the 24 copies displayed on the ferritin NP used in this study. It is also possible that the difference in immunogenicity could arise from the increased molecular weight of the BOAS NP immunogen compared to the BOAS alone, leading to a difference in moles of BOAS antigen in each cohort. However, due to the large size of the BOAS, the addition of the ferritin NP does not add a large amount of mass. 20 µg of BOAS NP or an 8mer BOAS equates to ~64 and ~83 µmoles of each HA component, respectively. This ~30% greater amount of HA in the 8mer BOAS, however, does not account for the observed difference in serum binding titers. Nevertheless, HA-specific responses were similar whether the BOAS were conjugated to the nanoparticle or not, indicating that HA proximity to the NP surface did not impact responses to each component. This observation is consistent with betacoronavirus quartet NPs as well. Additionally, BOAS conjugation to the NP significantly reduced the scaffold-directed response. The addition of the large BOAS projections to the NP surface likely masked the immunogenic scaffold epitopes (*Kraft et al., 2022*).

Collectively, this study demonstrates the versatility of the BOAS platform to present multiple HA subtypes as a single immunogen. This 'plug-and-play' approach can readily exchange HAs to elicit desired immune responses. BOAS are potentially advantageous over other multivalent display platforms, such as protein-based NPs, which can produce off-target responses due to their inherent immunogenicity (*Kanekiyo et al., 2013*). Furthermore, when genetic fusions of the antigen to nanoparticles is not possible, SpyTag-SpyCatcher (or another suitable conjugation approach) must be used, further contributing to scaffold-specific responses as well as additional multi-step manufacturing and purification challenges. Not only does our BOAS platform circumvent these potential caveats, but because this is a single polypeptide chain, this immunogen could readily be formulated as an mRNA lipid nanoparticle (LNP) (*Chaudhary et al., 2021*). The BOAS platform forms the basis for next-generation influenza vaccines and can more broadly be readily adapted to other viral antigens.

## Materials and methods

### Cell lines

Expi293F cells for protein expression were obtained from Thermo Fisher Scientific (A14527) and MDCK-SIAT1-PB1/H5/H7 cells for microneutralizations were kindly provided by Dr. Masaru Kanekiyo (NIAID/VRC). Cell lines tested negative for mycoplasma using MycoStrip mycoplasma detection kit (InvivoGen).

### BOAS immunogen and full-length soluble HA expression and purification

HA head for BOAS immunogens were designed based on the following sequences from the following subtypes: H1 (H1/A/Michigan/45/2015) (GenBank: AMA11475.1), H2 (H2/A/Japan/305/1957) (GenBank: AAA43185), H3 (H3/A/Kansas/14/2017) (GenBank: AVG71503), H4 (H4/A/American Black Duck/New Brunswick/00464/2010) (GenBank: AGG81749), H5 (H5/A/Viet Nam/1203/2004) (GenBank: ADD97095), H7 (H7/A/Shanghai/01/2014) (GenBank: AHK10800), H9 (H9/A/Guangdong/ MZ058/2016) (GenBank: AOR17625.1), and B (B/Malaysia/2506/2004) (GenBank: ABU99194). Head subdomains from these HAs were used in the BOAS immunogens, and full-length soluble ecto-domain (FLsE) trimers were used in ELISAs. Additional H1 (H1/A/Solomon Islands/3/2006) and H3 (H3/A/Hong Kong/1/1968) FLsEs were used in ELISAs as mismatched, antigenically distinct HAs for all BOAS. Sequences were codon optimized using IDT and subcloned into pVRC vectors for expression in mammalian cells. For BOAS immunogens, all HA heads were separated by a 'GA(GSS)$_4$AS' spacer. Specific head domain regions for each subtype were selected based on expression levels in expi293F cells. All protein sequences contained a C-terminal 8x-His tag for purification with an upstream 3C-protease cleavage site. Full-length HA trimers contained an additional C-terminal foldon (Fd) trimerization domain for expression as soluble trimers. BOAS immunogens and HA trimers were transfected in expi293F cells using Expifectamine transfection reagent and enhancers based on the manufacturer's instructions. Five days post-transfection, supernatant was harvested and clarified via centrifugation, then purified on a TALON cobalt resin via the 8x-His Tag. Recovered proteins were then purified on a Superdex 200 (S200) Increase 10/300 GL (for trimeric HAs) or Superose 6 Increase 10/300 GL (for BOAS) size-exclusion column in Dulbecco's Phosphate Buffered Saline (DPBS) within 48 hr of cobalt resin elution. Pooled fractions of BOAS were then treated with HRV-3C protease to remove tags for 16 hr at 4 °C, then passed over a TALON cobalt resin to remove any remaining protease, cleaved tags, and uncleaved protein. Trimeric HAs were used for ELISAs with purification tags present.

### IgG expression and purification

All antibody variable regions were codon optimized for mammalian cell expression and subcloned into pVRC vectors containing either heavy or light (kappa or lambda) with humanized constant regions. Equimolar heavy and light chain plasmids were co-transfected into HEK293F cells using polyethyleni-mine (PEI). Five days following transfection, supernatants were harvested, clarified via centrifugation, and purified on a Protein G resin. Purified monoclonal antibodies (mAbs) were then buffer exchanged into DPBS for use in ELISA assays.

### Nanoparticle assembly

C-terminally SpyTagged BOAS and N-terminal SpyCatcher nanoparticles were subcloned into pVRC expression vectors, then expressed in expi293F cells and purified via TALON cobalt resin as above. SpyTagged BOAS were further purified via size-exclusion chromatography on a Superose 6 column, as above, followed by cleavage of the C-terminal His tag via 3 C HRV protease to yield a C-terminal SpyTag. SpyCatcher nanoparticles were purified on a HiPrep Sephacryl S400 column in DPBS. Purified SpyCatcher nanoparticles was when mixed with an equimolar ratio mixture of each 4mer BOAS (H1, H2, H3, H4 and H5, H7, H9, B) at a 1.2 molar excess relative to the nanoparticle overnight at 4 °C. The mixture was then re-purified over an S400 column, concentrated, aliquoted, and snap-frozen in liquid nitrogen until use.

## Negative staining procedure for TEM

5 µl of the sample was adsorbed for 1 min onto a carbon-coated grid made hydrophilic by a 20 s exposure to a glow discharge (25 mA). Excess liquid was removed with filter paper (Whatman #1), the grid was then floated briefly on a drop of water (to wash away phosphate or salt), blotted again on a filter paper, and then stained with 0.75% uranyl formate or 1% uranyl acetate for 20–30 s. After removing the excess stain with a filter paper, the grids were examined in a JEOL 1200EX Transmission electron microscope or a Tecnai G² Spirit BioTWIN and images were recorded with an AMT 2 k CCD camera.

## Immunizations

C57BL/6 mice (Jackson Laboratory) (n=8 per group for 3-, 4-, 5-, 6-, 7-, and 8mer cohorts; n=5 for BOAS NP, NP, and mix cohorts) were immunized with 20 µg of BOAS immunogens of varying length and adjuvanted with 50% Sigmas Adjuvant for a total of 100 µL of inoculum. Immunogens and adjuvants were administered intramuscularly (IM) at day 0, day 14, and day 28. Serum samples were collected at days 0, 14, 28, and 42 (prior to immunogen administration). All experiments were conducted in 6–10 week old female mice under the institutional IACUC protocol 2014N000252.

## Serum ELISAs

High-binding 96-well plates (Corning) were coated with 200 ng per well of BOAS immunogen, nanoparticle, or trimeric HAs overnight at 4 °C in DPBS. The following day, plates were blocked with 1% BSA in PBS-T (DPBS + 0.1% Tween-20) for 1 hr at RT, rocking. Diluted serum in DPBS were generated starting at a 1:50 dilution, followed by 10-fold serial dilutions for a total of seven dilutions. After 1 hr, blocking buffer was discarded, and 40 µL of diluted serum was added and incubated for 1 hr at RT, rocking. Serum was then discarded, and plates were washed three times with PBS-T, after which 100 µL of HRP-conjugated anti-mouse secondary antibody (Abcam) at a 1:20,000 dilution was added and incubated for 1 hr at RT rocking. Following secondary antibody incubation, plates were again washed three times with PBS-T. Slow TMB development solution was then added to plates and incubated for 30–40 mins at RT, then stopped with an equal volume of 0.2 M sulfuric acid stop solution. Developed plates were then analyzed via absorbance measurements at 450 nm. Each condition was blank subtracted with a secondary-only control and plotted in GraphPad Prism 10 (v10.0.2). Serum endpoint titer (EPT) were determined using a non-linear regression (sigmoidal, four-parameter logistic (4PL) equation, where x is concentration) to determine the dilution at which dilution the blank-subtracted 450 nm absorbance value intersects a 0.1 threshold. Serum titers for individual mice against respective antigens are reported as log-transformed values of the EPT dilution.

## Serum competition ELISAs

Serum competition ELISAs were performed following a similar protocol to serum ELISAs described above. Following blocking with BSA in PBS-T, blocking solution was discarded and 40 µL of either DPBS (no competition control), a cocktail of humanized antibodies targeting the RBS and periphery 5J8 (*Hong et al., 2013*; *Krause et al., 2011*), 2G1 (*Krause et al., 2012*), K03.12 (*McCarthy et al., 2018*), H5.3 (*Winarski et al., 2015*), H7.167 (*Thornburg et al., 2016*), H1209 (*Bajic and Harrison, 2021*), a cocktail of humanized TI-directed antibodies S5V2-29 (*Watanabe et al., 2019*), D1 H1-17/H3-14 (*McCarthy et al., 2021*), D2 H1-1/H3-1 (*McCarthy et al., 2021*), or a negative control antibody MEDI8852 (*Kallewaard et al., 2016*) were added at a concentration of 100 µg/mL per antibody. Plates were incubated with competing antibodies or controls for 1 hr at RT. Serum from mice from respective BOAS cohorts were diluted by 1:5000 (3mer and 6mer) or 1:10,000 (4mer, 5mer, 7mer, and 8mer) in DPBS, then 40 µL added on top of competing antibody for a final dilution of 1:10,000 (3mer and 6mer) or 1:20,000 (4mer, 5mer, 7mer, and 8mer). These dilutions were selected based on a dilution range in serum ELISAs in which serum reactivity behaved linearly. Plates were incubated with serum and competing antibodies for an additional 1 hr at RT, washed three times with PBS-T, then incubated with an HRP-conjugated human/bovine/horse cross-adsorbed anti-mouse secondary antibody (Southern Biotech) at a 1:20,000 dilution for 1 hr at RT. The remaining wash and development steps were performed as in the serum ELISA.

## Microneutralization assays

Serum microneutralization assays using R3 viruses followed the published protocol in *Creanga et al., 2021*. Briefly, MDCK-SIAT1-PB1/H5/H7 cells were thawed into D10 media with penicillin and streptomycin (P/S) and 10% heat-inactivated fetal bovine serum (FBS) three days prior to infection. Cells were seeded two days later at $1.5 \times \times 10^4$ cells/well in flu media (Opti-MEM +0.01% FBS+P/S+0.3% Bovine Serum Albumin (BSA) +Ca/Mg) overnight at 37 °C. Serum was also treated with receptor destroying enzyme (RDE) II at a 1:3 dilution overnight at 37 °C, followed by inactivation at 56 °C for 30–60 min. Treated serum was then diluted in media at 1:3 for a final starting dilution of 1:32. On the day of infection, cells were treated with 60 µL of virus in flu media + TPCK-trypsin for 1 hr at 37 °C with 5% $CO_2$. After 1 hr, media was removed via aspiration and cells were treated with 100 µL treated serum. The following day, fluorescent foci were counted on a Zeiss Celldiscoverer 7+LSM900+Airyscan 2 (CD7 +LSM900) to determine neutralization. Counts were normalized to a cell control and untreated virus control, and IC50 values for neutralization were determined via a fit to a 4PL non-linear regression in technical duplicate in Prism v10.0.2 (GraphPad).

## Homology analyses

Homology analyses on BOAS were conducted using a manual method analogous to Consurf (*Glaser et al., 2003*; *Ashkenazy et al., 2016*). HA head domain structures (or closely related variants) were aligned in Pymol to the H2 JP-57 structure (PDB: 2WRE). Structures used for alignments were as follows: H1 (PDB: 6XGC), H3 (PDB: 4O5N), H4 (PDB: 5XL3), H5 (PDB: 6CFG), H7 (PDB: 6FYU), H9 (PDB: 1JSI), B (PDB: 4FQJ). Any mismatches in specific amino acids were adjusted using the mutagenesis wizard in Pymol. Following structural alignment, each position on the HA head domain was scored for % aa identity at each position on a scale of 1–9, where 1=10–19% identical, 2=20–29% identical … to 9=90–100% identical, which we refer to as a 'conservation score.' This was repeated for each composition of BOAS (i.e. 3-, 4-, 5-, … 8mer). Each position on the H2 structure was then pseudo-colored based on the conservation score. Average amino acid conservation scores were defined as the average of conservation score over the whole sequence (overall), or of amino acids within a given epitope. TI epitope amino acids were defined as residues 60, 86–106, and 212–232 (H1 numbering), and RBS epitope amino acids were defined as residues 94–96, 128–146, 153–161, 182–194, and 222–229 (H1 numbering).

## Statistical analysis

Significance for ELISAs and microneutralization assays were determined using Prism (GraphPad Prism v10.2.3). ELISAs comparing serum reactivity and microneutralization and comparing >2 samples were analyzed using a Kruskal-Wallis test with Dunn's post-hoc test to correct for multiple comparisons. Multiple comparisons were made between each possible combination or relative to a control group, where indicated. ELISAs comparing two samples were analyzed using a Mann-Whitney test. Significance was assigned with the following: *=$p<0.05$, **=$p<0.01$, ***=$p<0.001$, and ****=$p<0.0001$. Where conditions are compared and no significance is reported, the difference was non-significant.

# Acknowledgements

We thank the mouse facility at the Ragon Institute for mouse maintenance. We also thank Nicholas Lamson for helping with DLS measurements for nanoparticle characterization. We would also like to thank Maria Ericcson for acquiring electron microscopy images. We acknowledge support from R01 AI146779 and P01 AI089618 (AGS) and R21AI193280, R01AI137057, R01AI153098, and R01AI155447 (DL). This research has been funded in whole or in part with federal funds under a contract from the National Institute of Allergy and Infectious Diseases, NIH contract 75N93019C00050 (AGS).

# Additional information

#### Competing interests

Dana Thornlow Lamson, Aaron G Schmidt: Has filed a provisional patent covering the work described here: US Patent Application number 63/547,974. The other authors declare that no competing interests exist.

## Funding

| Funder | Grant reference number | Author |
|---|---|---|
| National Institute of Allergy and Infectious Diseases | R01 AI146779 | Aaron G Schmidt |
| National Institute of Allergy and Infectious Diseases | P01 AI089618 | Aaron G Schmidt |
| National Institute of Allergy and Infectious Diseases | R21AI193280 | Daniel Lingwood |
| National Institute of Allergy and Infectious Diseases | R01AI137057 | Daniel Lingwood |
| National Institute of Allergy and Infectious Diseases | R01AI153098 | Daniel Lingwood |
| National Institute of Allergy and Infectious Diseases | R01AI155447 | Daniel Lingwood |
| National Institute of Allergy and Infectious Diseases | 75N93019C00050 | Aaron G Schmidt |

The funders had no role in study design, data collection and interpretation, or the decision to submit the work for publication.

## Author contributions

Dana Thornlow Lamson, Conceptualization, Data curation, Formal analysis, Validation, Investigation, Writing – original draft, Writing – review and editing; Faez Amokrane Nait Mohamed, Mya Vu, Daniel P Maurer, Larance Ronsard, Data curation, Writing – review and editing; Daniel Lingwood, Funding acquisition, Writing – review and editing; Aaron G Schmidt, Conceptualization, Supervision, Funding acquisition, Project administration, Writing – review and editing

## Author ORCIDs

Dana Thornlow Lamson ⓘ https://orcid.org/0000-0001-9440-9656
Faez Amokrane Nait Mohamed ⓘ https://orcid.org/0000-0002-5239-4238
Larance Ronsard ⓘ https://orcid.org/0000-0002-5307-1940
Daniel Lingwood ⓘ https://orcid.org/0000-0001-5631-9238
Aaron G Schmidt ⓘ https://orcid.org/0000-0003-3627-2553

## Ethics

All experiments were conducted in 6-10 week old female mice under the institutional IACUC protocol (2014N000252).

Reviewer #2 (Public review): https://doi.org/10.7554/eLife.97364.3.sa1
Reviewer #3 (Public review): https://doi.org/10.7554/eLife.97364.3.sa2
Author response https://doi.org/10.7554/eLife.97364.3.sa3

# Additional files

## Supplementary files

MDAR checklist

## Data availability

All data generated or analyzed during this study are included in the manuscript, figures, figure supplements, and source data.

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
