## [Editor Report · eLife Assessment]

This **valuable** manuscript describes the immunogenicity of a bead-on-a-string immunogen that allows the inclusion of multiple HA subtypes. The evidence to support the claims is **convincing**, and more importantly, this approach could be adapted to other vaccine platforms.

---

## [Referee Report · Reviewer #2 (Public review)]

Summary:

The authors describe a "beads-on-a-string" (BOAS) immunogen, where they link, using a non-flexible glycine linker, up to eight distinct hemagglutinin (HA) head domains from circulating and non-circulating influenzas and assess their immunogenicity. They also display some of their immunogens on ferritin NP and compare the immunogenicity. They conclude that this new platform can be useful to elicit robust immune responses to multiple influenza subtypes using one immunogen and that it can also be used for other viral proteins.

Strengths:

The paper is clearly written. While the use of flexible linkers has been used many times, this particular approach (linking different HA subtypes in the same construct resembling adding beads on a string, as the authors describe their display platform) is novel and could be of interest.

Comments on revisions:

The authors have addressed most comments. Some mistakes/issues remain:

TI should be defined earlier on line 61 not on line 196

No legend for Figure 3E - it looks like this is where the authors did the first immunization with the "mix" to compare to the BOAs but strangely they do not mention this in the response to reviewers letter and only mention fig 6G and 7

Maybe add "mix" to the title of Figure 3?

In Figure 6G they do show the response to the mix but do not mention it in the immunizations for that figure. Also weird because obviously the mix is not a NP while this figure addresses NP format.

Line 796 - pseudo viruses

The authors should add some clarification in the paper as they did in response to reviewers.

---

## [Referee Report · Reviewer #3 (Public review)]

This work describes the tandem linkage of influenza hemagglutinin (HA) receptor binding domains of diverse subtypes to create 'beads on a string' (BOAS) immunogens. They show that these immunogens elicit ELISA binding titers against full-length HA trimers in mice, as well as varying degrees of vaccine mismatched responses and neutralization titers. They also compare these to BOAS conjugated on ferritin nanoparticles and find that this did not largely improve immune responses. This work offers a new type of vaccine platform for influenza vaccines, and this could be useful for further studies on the effects of conformation and immunodominance on the resulting immune response.

Overall, the central claims of immunogenicity in a murine model of the BOAS immunogens described here are supported by the data.

Strengths included the adaptability of the approach to include several, diverse subtypes of HAs. The determination of an optimal composition of strains in the 5-BOAS that overall yielded the best immune responses was an interesting finding and one that could also be adapted to other vaccine platforms. Lastly, as the authors discuss, the ease of translation to an mRNA vaccine is indeed a strength of this platform.

One interesting and counter-intuitive result is the high levels of neutralization titers seen to vaccine-mismatched, group 2 H7 in the 5-BOAS group that differs from the 4-BOAS with the addition of a group 1 H5 RBD. At the same time, no H5 neutralization titers were observed for any of the BOAS immunogens, yet they were seen for the BOAS-NP. Uncovering where these immune responses are being directed and why these discrepancies are being observed would be informative future work.

There are a few caveats in the data that should be noted:

(1) 20 ug is a pretty high dose for a mouse and the majority of the serology presented is after 3 doses at 20 ug. By comparison, 0.5-5 ug is a more typical range (https://www.ncbi.nlm.nih.gov/pmc/articles/PMC6380945/, https://www.ncbi.nlm.nih.gov/pmc/articles/PMC9980174/). Also, the authors state that 20 ug per immunogen was used, including for the BOAS-NP group, which would mean that the BOAS-NP group was given a lower gram dose of HA RBD relative to the BOAS groups.

(2) Serum was pooled from all animals per group for neutralization assays, instead of testing individual animals. This could mean that a single animal with higher immune responses than the rest in the group could dominate the signal and potentially skew the interpretation of this data.

(3) In Figure S2, it looks like an apparent increase in MW by changing the order of strains here, which may be due to differences in glycosylation. Further analysis would be needed to determine if there are discrepancies in glycosylation amongst the BOAS immunogens and how those differ from native HAs.

Comments on revisions:

The authors have addressed all concerns upon revision.

---

## [Author Response]

The following is the authors’ response to the original reviews

**Public Reviews:**

**Reviewer #1 (Public Review):**
Summary:In this manuscript by Thronlow Lamson et al., the authors develop a "beads-on-a-string" or BOAS strategy to link diverse hemagglutinin head domains, to elicit broadly protective antibody responses. The authors are able to generate varying formulations and lengths of the BOAS and immunization of mice shows induction of antibodies against a broad range of influenza subtypes. However, several major concerns are raised, including the stability of the BOAS, that only 3 mice were used for most immunization experiments, and that important controls and analyses related to how the BOAS alone, and not the inclusion of diverse heads, impacts humoral immunity.Strengths:Vaccine strategy is new and exciting.Analyses were performed to support conclusions and improve paper quality.Weaknesses:Controls for how different hemagglutinin heads impact immunity versus the multivalency of the BOAS.Only 3 mice were used for most experiments.There were limited details on size exclusion data.

We appreciate the reviewer’s comments and have made the following changes to the manuscript.

(1) We recognize that deconvoluting the effect of including a diverse set of HA heads and multivalency in the BOAS immunogens is necessary to understand the impact on antigenicity. Therefore, we now include a cocktail of the identical eight HA heads used in the 8-mer and BOAS nanoparticle (NP) as an additional control group. While we observed similar HA binding titers relative to the 8-mer and BOAS NP groups, the cocktail group-elicited sera was unable to neutralize any of the viruses tested; multivalency thus appears to be important for eliciting neutralizing responses

(2) We increased the sample size by repeated immunizations with n=5 mice, for a total of n=8 mice across two independent experiments.

(3) We expanded the details on size exclusion data to include:

a) extended chromatograms from Figure 2C as Supplemental Figure 3.

b) additional details in the materials and methods section (lines 370-372):

“Recovered proteins were then purified on a Superdex 200 (S200) Increase 10/300 GL (for trimeric HAs) or Superose 6 Increase 10/300 GL (for BOAS) size-exclusion column in Dulbecco’s Phosphate Buffered Saline (DPBS) within 48 hours of cobalt resin elution.”

**Reviewer #2 (Public Review):**
Summary:The authors describe a "beads-on-a-string" (BOAS) immunogen, where they link, using a non-flexible glycine linker, up to eight distinct hemagglutinin (HA) head domains from circulating and non-circulating influenzas and assess their immunogenicity. They also display some of their immunogens on ferritin NP and compare the immunogenicity. They conclude that this new platform can be useful to elicit robust immune responses to multiple influenza subtypes using one immunogen and that it can also be used for other viral proteins.Strengths:The paper is clearly written. While the use of flexible linkers has been used many times, this particular approach (linking different HA subtypes in the same construct resembling adding beads on a string, as the authors describe their display platform) is novel and could be of interest.Weaknesses:The authors did not compare to individuals HA ionized as cocktails and did not compare to other mosaic NP published earlier. It is thus difficult to assess how their BOAS compare.Other weaknesses include the rationale as to why these subtypes were chosen and also an explanation of why there are different sizes of the HA1 construct (apart from expression). Have the authors tried other lengths? Have they expressed all of them as FL HA1?

We appreciate the reviewer’s comments. We responded to the concerns below and modified the manuscript accordingly.

(1) We recognize that including a “cocktail” control is important to understand how the multivalency present in a single immunogen affects the immune response. We now include an additional control group comprised of a mixture of the same eight HA heads used in the 8-mer and the BOAS nanoparticle (NP). While this cocktail elicited similar HA binding titers relative to the 8-mer and BOAS NP immunogens (Fig. 6G), there was no detectable neutralization any of the viruses tested (Fig. 7).

(2) In the introduction we reference other multivalent display platforms but acknowledge that distinct differences in their immunogen design platforms make direct comparisons to ours difficult—which is ultimately why we did not use them as comparators for our in vivo studies. Perhaps most directly relevant to our BOAS platform is the mosaic HA NP from Kanekiyo *et al*. (PMID 30742080). Here, HA heads, with similar boundaries to ours, were selected from historical H1N1 strains. These NPs however were significantly less antigenic diverse relative to our BOAS NPs as they did not include any group 2 (e.g.*,* H7, H9) or B influenza HAs; restricting their multivalent display to group 1 H1N1s likely was an important factor in how they were able to achieve broad, neutralizing H1N1 responses. Additionally, Cohen et al. (PMID 33661993) used similarly antigenically distinct HAs in their mosaic NP, though these included full-length HAs with the conserved stem region, which likely has a significant impact on the elicited cross-reactive responses observed. Lastly, we reference Hills *et al*. (PMID 38710880), where authors designed similar NPs with four tandemly-linked betacoronoavirus receptor binding domains (RBDs) to make “quartets”. In contrast to our observations, the authors observed increased binding and neutralization titers following conjugation to protein-based NPs. We acknowledge potential differences between the studies, such as the antigen and larger VLP NP, that could lead to the different observed outcomes.

(3) We intended to highlight the “plug-and-play” nature of the BOAS platform; theoretically any HA subtype could be interchanged into the BOAS. To that end, our rationale for selecting the HA subtypes in our proof-of-principle immunogen was to include an antigenically diverse set of circulating and non-circulating HAs that we could ultimately characterize with previously published subtype-specific antibodies that were also conformation-specific. In doing so, these diagnostic antibodies could confirm presence and conformation integrity of each component. We intentionally did not include HA subtypes that we did not have a conformation-specific antibody for.

The different sizes of HA head domains was determined exclusively by expression of the recombinant protein. We have not attempted expression of full-length HA1 domains. Furthermore, we have not attempted to express the full-length HA (inclusive of HA1 and HA2) in our BOAS platform. The primary reason was to avoid including the conserved stem region of HA2 which may distract from the HA1 epitopes (e.g.*,* receptor binding site, lateral patch) that can be engaged by broadly neutralizing antibodies. Additionally, the full-length HA is inherently trimeric and may not be as amenable to our BOAS platform as the monomeric HA1 head domain.

**Reviewer #3 (Public Review):**
This work describes the tandem linkage of influenza hemagglutinin (HA) receptor binding domains of diverse subtypes to create 'beads on a string' (BOAS) immunogens. They show that these immunogens elicit ELISA binding titers against full-length HA trimers in mice, as well as varying degrees of vaccine mismatched responses and neutralization titers. They also compare these to BOAS conjugated on ferritin nanoparticles and find that this did not largely improve immune responses. This work offers a new type of vaccine platform for influenza vaccines, and this could be useful for further studies on the effects of conformation and immunodominance on the resulting immune response.Overall, the central claims of immunogenicity in a murine model of the BOAS immunogens described here are supported by the data.Strengths included the adaptability of the approach to include several, diverse subtypes of HAs. The determination of the optimal composition of strains in the 5-BOAS that overall yielded the best immune responses was an interesting finding and one that could also be adapted to other vaccine platforms. Lastly, as the authors discuss, the ease of translation to an mRNA vaccine is indeed a strength of this platform.One interesting and counter-intuitive result is the high levels of neutralization titers seen in vaccine-mismatched, group 2 H7 in the 5-BOAS group that differs from the 4-BOAS with the addition of a group 1 H5 RBD. At the same time, no H5 neutralization titers were observed for any of the BOAS immunogens, yet they were seen for the BOAS-NP. Uncovering where these immune responses are being directed and why these discrepancies are being observed would constitute informative future work.There are a few caveats in the data that should be noted:(1) 20 ug is a pretty high dose for a mouse and the majority of the serology presented is after 3 doses at 20 ug. By comparison, 0.5-5 ug is a more typical range (https://www.ncbi.nlm.nih.gov/pmc/articles/PMC6380945/, https://www.ncbi.nlm.nih.gov/pmc/articles/PMC9980174/). Also, the authors state that 20 ug per immunogen was used, including for the BOAS-NP group, which would mean that the BOAS-NP group was given a lower gram dose of HA RBD relative to the BOAS groups.

We agree that this is on the “upper end” of recombinant protein dose. While we did not do a dose-response, we now include serum analyses after a single prime. The overall trends and reactivity to matched and mis-matched BOAS components remained similar across days d28 and d42. However, the differences between the BOAS and BOAS NP groups and the mixture group were more pronounced at d28, which reinforces our observation that the multivalency of the HA heads is necessary for eliciting robust serum responses to each component. These data are included in Supplemental Figure 5, and we’ve modified the text (lines 185-187) to include;

“Similar binding trends were also observed with d28 serum, though the difference between the 8mer and mix groups was more pronounced at d28 (Supplemental Figure 5).”

Additionally, we acknowledge that there is a size discrepancy between the BOAS NP and the largest BOAS, leading to an approximately ~15-fold difference on a per mole basis of the BOAS immunogen. The smallest and largest BOAS also differ by ~ 2.5-fold on a per mole basis; this could favor the overall amount of the smaller immunogens, however because vaccine doses are typically calculated on a mg per kg basis, we did not calculate on a molar basis for this study. Any promising immunogens will be evaluated in dose-response study to optimize elicited responses.

(2) Serum was pooled from all animals per group for neutralization assays, instead of testing individual animals. This could mean that a single animal with higher immune responses than the rest in the group could dominate the signal and potentially skew the interpretation of this data.

We repeated the neutralization assays with data points for individual mice. There does appear to be variability in the immune response between mice. This is most noticeable for responses to the H5 component. We are currently assessing what properties of our BOAS immunogen might contribute to the variability across individual mice.

(3) In Figure S2, it looks like an apparent increase in MW by changing the order of strains here, which may be due to differences in glycosylation. Further analysis would be needed to determine if there are discrepancies in glycosylation amongst the BOAS immunogens and how those differ from native HAs.

There does appear to be a relatively small difference in MW between the two BOAS configurations shown in **Figure S2**. This could be due to differences in glycosylation, as the reviewer points out, and in future studies, we intend to assess the influence of native glycosylation on antibody responses elicited by our BOAS immunogens.

**Recommendations for the authors:**

**Reviewer #1 (Recommendations For The Authors):**
Major Concerns(1) From Figure 2D-E, it looks like BOAS are forming clusters, rather than a straight line. Do these form aggregates over time? Both at 4 degrees over a few days or after freeze-thaw cycle(s)? It is unclear from the SEC methods how long after purification this was performed and stability should be considered.

Due to the inherent flexibility of the Gly-Ser linker between each component we do not anticipate that any rigidity would be imposed resulting in a “straight line”. Nevertheless, we appreciate the reviewers concern about the long-term stability of the BOAS immunogens. To address this, we include (1) the extended chromatograms from Figure 2C as Supplemental Figure 3 to show any aggregates present, (2) traces from up to 48 hours post-IMAC, and (3) chromatograms following a freeze-thaw cycle. Post-IMAC purification there is a minor (<10% total peak height) at ~9mL corresponding to aggregation. Note, we excluded this aggregation for immunizations. Post freeze-thaw cycle, we can see that upon immediate (<24hrs) thawing, the BOAS maintain a homogeneous peak with no significant (<10%) aggregation or degradation peak. However, after ~1 week post-freeze-thaw cycle at 4C, additional peaks within the chromatogram correspond to degradation of the BOAS.

We modified the materials and methods section to state (lines 370-372)

“Recovered proteins were then purified on a Superdex 200 (S200) Increase 10/300 GL (for trimeric HAs) or Superose 6 Increase 10/300 GL (for BOAS) size-exclusion column in Dulbecco’s Phosphate Buffered Saline (DPBS) within 48 hours of cobalt resin elution.”

We commented on BOAS stability in the results section (lines 142-148)

“Following SEC, affinity tags were removed with HRV-3C protease; cleaved tags, uncleaved BOAS, and His-tagged enzyme were removed using cobalt affinity resin and snap frozen in liquid nitrogen before immunizations. BOAS maintained monodispersity upon thawing, though over time, degradation was observed following longer term (>1 week) storage at 4C (Supplemental Figure 3). This degradation became more significant as BOAS increased in length (Supplemental Figure 3).”

We also included in the discussion (lines 277-279):

“Notably, for longer BOAS we observed degradation following longer term storage at 4C, which may reflect their overall stability.”

(2) Figures 3-4 and 6-7, to make conclusions off of 3 mice per group is inappropriate. A sample size calculation should have been conducted and the appropriate number of mice tested. In addition, two independent mouse experiments should always be performed. Moreover, the reliability of the statistical tests performed seems unlikely, given the very small sample size.

We agree that additional mice are necessary to make assessments regarding immunogenicity and cross-reactivity differences between the immunogens. To address this, we repeated the immunization with 5 additional mice, for a total of n=8 mice over two independent experiments. We incorporated these data into Figure 3B-D, as well as an additional Figure 3E (see below). We also now report the log-transformed endpoint titer (EPT) values rather than reciprocal EC50 values and added clarity to statistical analyses used. We have added the following lines to the methods section

lines 427-431:

“Serum endpoint titer (EPT) were determined using a non-linear regression (sigmoidal, four-parameter logistic (4PL) equation, where x is concentration) to determine the dilution at which dilution the blank-subtracted 450nm absorbance value intersect a 0.1 threshold. Serum titers for individual mice against respective antigens are reported as log transformed values of the EPT dilution.”

lines 406-408:

“C57BL/6 mice (Jackson Laboratory) (n=8 per group for 3-, 4-, 5-, 6-, 7-, and 8mer cohorts; n=5 for BOAS NP, NP, and mix cohorts) were immunized with 20µg of BOAS immunogens of varying length and adjuvanted with 50% Sigmas Adjuvant for a total of 100µL of inoculum.”

lines 482-490:

“Statistical Analysis

Significance for ELISAs and microneutralization assays were determined using Prism (GraphPad Prism v10.2.3). ELISAs comparing serum reactivity and microneutralization and comparing >2 samples were analyzed using a Kruskal-Wallis test with Dunn’s post-hoc test to correct for multiple comparisons. Multiple comparisons were made between each possible combination or relative to a control group, where indicated. ELISAs comparing two samples were analyzed using a Mann-Whitney test. Significance was assigned with the following: * = p<0.05, ** = p<0.01, *** = p<0.001, and **** = p<0.0001. Where conditions are compared and no significance is reported, the difference was non-significant.”

(3) One critical control that is missing is a homogenous BOAS, for example, just linking one H1 on a BOAS. Does oligomerization and increasing avidity alone improve humoral immunity?

We agree that this is an interesting point, However, to address the impact of oligomerization and avidity on humoral immunity, we now include an additional control with a cocktail of HA heads used in the 8mer. We have incorporated this into Figure 3A, 3D and 3E, Figure 6G, and Figure 7.

Additionally, we have added the following lines in the manuscript:

lines 38-40:

“Finally, vaccination with a mixture of the same HA head domains is not sufficient to elicit the same neutralization profile as the BOAS immunogens or nanoparticles.”

lines 105-106:

“Additionally, we showed that a mixture of the same HA head components was not sufficient to recapitulate the neutralizing responses elicited by the BOAS or BOAS NP.”

lines 169-172:

“To determine immunogenicity of each BOAS immunogen, we performed a prime-boost-boost vaccination regimen in C5BL/6 mice at two-week intervals with 20µg of immunogen and adjuvanted with Sigma Adjuvant (Figure 3A). We compared these BOAS to a control group immunized with a mixture of the eight HA heads present in the 8mer.”

lines 265-267:

“There were qualitatively immunodominant HAs, notably H4 and H9, and these were relatively consistent across BOAS in which they were a component. This effect was reduced in the mix cohort.”

(4) While some cross-reactivity is likely (Figure 6G), there is considerable loss of binding when there is a mismatch. Of the antibodies induced, how much of this is strain-specific? For example, how well do serum antibodies bind to a pre-2009 H1?

We agree with the reviewer that there is a considerable loss of binding when there is a mismatched HA component. To better understand this and incorporate a mismatched strain into our analysis of the 8mer and BOAS NP, we looked at serum binding titers to a pre-2009 H1, H1/Solomon Islands/2006, and an antigenically distinct H3, H3/Hong Kong/1968. We have incorporated this data into Figures 3D, 3E, 6F and 6G. We observed relatively high titers against both a mismatched H1 and H3, indicating that the BOAS maintain high titers against subtype-specific strains that are conserved over considerable antigenic distance. However, this was similar in the mixture group, indicating that this may not be specific to oligomerization of BOAS immunogens.

We added the following to the methods section:

lines 357-361

“Head subdomains from these HAs were used in the BOAS immunogens, and full-length soluble ectodomain (FLsE) trimers were used in ELISAs. Additional H1 (H1/A/Solomon Islands/3/2006) and H3 (H3/A/Hong Kong/1/1968) FLsEs were used in ELISAs as mismatched, antigenically distinct HAs for all BOAS.”

Minor Concerns(1) Line 44-46, the deaths per year are almost exclusively due to seasonal influenza outbreaks caused by antigenically drifted viruses in humans, not those spilling over from avian sp. and swine. For accuracy, please adjust this sentence.

We have adjusted lines 45-48 to say “This is largely a consequence of viral evolution and antigenic drift as it circulates seasonally within humans and ultimately impacts vaccine effectiveness. Additionally, the chance for spillover events from animal reservoirs (e.g., avian, swine) is increasing as population and connectivity also increase.”

(2) Figure 4D-E, provide a legend for what the symbols indicate, or simply just put the symbol next to either the homology score and % serum competition labels on the y-axis.

We have included a legend in Figures 4D,E to distinguish between homology score and % serum competition

(3) I am a bit confused by the data presented in Figure 7. The figure legend says the two symbols represent technical replicates. How? Is one technical replicate of all the mice in a group averaged and that's what's graphed? If so, this is not standard practice. I would encourage the authors to show the average technical replicates of each animal, which is standard.

We thank the reviewer for their suggestion, and we have revised Figure 7 such that each symbol represents a single animal for n=5 animals. We have also adjusted the figure caption to the following:

“Figure 7: Microneutralization titers to matched and mis-matched virus- Microneutralization of matched and mis-matched psuedoviruses: H1N1 (green, top left), H3N2 (orange, top right), H5N1 (yellow, bottom left), and H7N9 viruses (pink, bottom right) with d42 serum. Solid bars below each plot indicate a matched sub-type, and striped bars indicate a mis-matched subtype (i.e. not present in the BOAS). NP negative controls were used to determine threshold for neutralization. Upper and lower dashed lines represent the first dilution (1:32) (for H1N1, H3N2, and H5N1) or neutralization average with negative control NP serum (H7N9), and the last serum dilution (1:32,768), respectively, and points at the dashed lines indicate IC50s at or outside the limit of detection. Individual points indicate IC50 values from individual mice from each cohort (n=5). The mean is denoted by a bar and error bars are +/- 1 s.d., * = p<0.05 as determined by a Kruskal-Wallis test with Dunn’s multiple comparison post hoc test relative to the mix group.”

(4) Paragraphs 298-313, multiple studies are referred to but not referenced.

We have added the following references to this section:

(38) Kanekiyo, M. et al. Self-assembling influenza nanoparticle vaccines elicit broadly neutralizing H1N1 antibodies. Nature 498, 102–106 (2013).

(48) Hills, R. A. et al. Proactive vaccination using multiviral Quartet Nanocages to elicit broad anti-coronavirus responses. Nat. Nanotechnol. 1–8 (2024) doi:10.1038/s41565-024-01655-9.

(65) Jardine, J. et al. Rational HIV immunogen design to target specific germline B cell receptors. Science 340, 711–716 (2013).

(66) Tokatlian, T. et al. Innate immune recognition of glycans targets HIV nanoparticle immunogens to germinal centers. Science 363, 649–654 (2019).

(67) Kato, Y. et al. Multifaceted Effects of Antigen Valency on B Cell Response Composition and Differentiation In Vivo. Immunity 53, 548-563.e8 (2020).

(68) Marcandalli, J. et al. Induction of Potent Neutralizing Antibody Responses by a Designed Protein Nanoparticle Vaccine for Respiratory Syncytial Virus. Cell 176, 1420-1431.e17 (2019).

(69) Bruun, T. U. J., Andersson, A.-M. C., Draper, S. J. & Howarth, M. Engineering a Rugged Nanoscaffold To Enhance Plug-and-Display Vaccination. ACS Nano 12, 8855–8866 (2018).

(70) Kraft, J. C. et al. Antigen- and scaffold-specific antibody responses to protein nanoparticle immunogens. Cell Reports Medicine 100780 (2022) doi:10.1016/j.xcrm.2022.100780.

**Reviewer #2 (Recommendations For The Authors):**
Can the authors define "detectable titers"?Maybe add a threshold value of reciprocal EC on the figure for each plot.

We recognize the reviewers concern with reporting serum titers in this way, and we have adjusted our reported titers as endpoint titers (EPT) with a dotted line for the first detectable dilution (1:50). We have also adjusted the methods section to reflect this change:

(lines 427-431)

“Serum endpoint titer (EPT) were determined using a non-linear regression (sigmoidal, four-parameter logistic (4PL) equation, where x is concentration) to determine the dilution at which dilution the blank-subtracted 450nm absorbance value intersect a 0.1 threshold. Serum titers for individual mice against respective antigens are reported as log transformed values of the EPT dilution.”

It also appears that not all X-mer elicits an immune response against matched HA, e.g. for the 7 and 8 -mer. Not sure why the authors do not mention this. It could be due to too many HAs, not sure.

We apologize for the confusion, and agree that our original method of reporting EC50 values does not reflect weak but present binding titers. Upon further analysis with additional mice as well as adjusting our method of reporting titers, it is easier to see in Figure 3D that all X-mer BOAS do indeed elicit binding detectable titers to matched HA components.

It will be nice to add a conclusion to the cross-reactivity - again it appears that past 6-mer there has been a loss in cross-reactivity even though there are more subtypes on the BOAS.Also, the TI seemed to be the more conserved epitope targeted here.(Of note these two are mentioned in the discussion)

We have updated the results section to include the following:

(lines 281-294)

“Based on the immunogenicity of the various BOAS and their ability to elicit neutralizing responses, it may not be necessary to maximize the number of HA heads into a single immunogen. Indeed, it qualitatively appears that the intermediate 4-, 5-, and 6mer BOAS were the most immunogenic and this length may be sufficient to effectively engage and crosslink BCR for potent stimulation. These BOAS also had similar or improved binding cross-reactivity to mis-matched HAs as compared to longer 7- or 8mer BOAS. Notably, the 3mer BOAS elicited detectable cross-reactive binding titers to H4 and H5 mismatched HAs in all mice. This observed cross-reactivity could be due to sequence conservation between the HAs, as H3 and H4 share ~51% sequence identity, and H1 and H2 share ~46% and ~62% overall sequence identity with H5, respectively (Supplemental Figure 6). Additionally, the degree of surface conservation decreased considerably beyond the 5mer as more antigenically distinct HAs were added to the BOAS. These data suggest that both antigenic distance between HA components and BOAS length play a key role in eliciting cross-reactive antibody responses, and further studies are necessary to optimize BOAS valency and antigenic distance for a desired response.”

Figure 5E, the authors could indicate which subtype each mab is specific to for those who are not HA experts. (They have them color-coded but it is hard to see because very small).The authors also do not explain why 3E5 does not bind well to H1, H2, H3, H4 4-mer BOA, etc...

We apologize for the lack of clarity in this figure. We updated Figure 5E to include the subtype it is specific for as well as listing the antibodies and their subtype and targeted epitope in the figure caption.

MinorFigure 1B zoom looks like the line is hidden to the structure - should come in front

We adjusted the figure accordingly.

Line 127 - whether the order

Corrected

What is the rationale for thinking that a different order will lead to a different expression and antigenic results?

We thank the reviewer for this question. We did not necessarily anticipate a difference in protein expression based on BOAS order We, however, wanted to verify that our platform was indeed “plug-and-play” platform and we could readily exchange components and order. We do, however, hypothesize that a different order may in fact lead to different antigenic results. We think that the conformation of the BOAS as well as physical and antigenic distance of HA components may influence cross-linking efficiency of BCRs and lead to different antigenic results with different levels of cross-reactivity. For example, a BOAS design with a cluster of group 1 HAs followed by a cluster of group 2 HAs, rather than our roughly alternating pattern could impact which HAs are in proximity to each other or could be potentially shielded in certain conformations, and thus could affect antigenic results. We expand on this rationale in the discussion in lines 310-314:

“Further studies with different combinations of HAs could aid in understanding how length and composition influences epitope focusing. For example, a BOAS design with a cluster of group 1 HAs followed by a cluster of group 2 HAs, rather than our roughly alternating pattern could impact which HAs are in close proximity to one other or could be potentially shielded in certain conformations, and thus could affect antigenic results.”

Maybe list HA#1 HA#2 HA#3 instead of HA1, HA2, HA3 to make sure it is not confounded with HA2 and HA2

We agree that this may be confusing for readers, and have adjusted Figure 1C to show HA#1, HA#2, etc.

For nsEM, do the authors have 2D classes and even 3D reconstructions? Line 148-149: maybe or just because there are more HAs.

We did not obtain 2D class or 3D reconstructions of these BOAS. However, we do agree with the reviewer that the collapsed/rosette structure of the 8mer BOAS may be a consequence of the additional HA heads as well as the flexible Gly-Ser linkers between the components. We have added clarify to our statement in the discussion to read:

lines 154-156:

“This is likely a consequence of the flexible GSS linker separating the individual HA head components as well as the addition of significantly more HA head components to the construct.”.

Line 153 " interface-directed" - what does this mean?

We apologize for any confusion- we intend for “interface-directed” to refer antibodies that engage the trimer interface (TI) epitope between HA protomers. We have adjusted the manuscript to use the same terminology throughout, i.e. trimer interface or its abbreviation, TI.

For Figure 2 F - do you have a negative control? Usually one does not determine an ELISA KD, it is not very accurate but shows binding in terms of OD value.

We did include a negative control, MEDI8852, a stem-directed antibody, though it was not shown in the figure because we observed no binding, as expected. This negative control antibody was also used in Figure 5E for characterizing the BOAS NPs, and also shows no binding. We recognize that in an ELISA the KD is an equilibrium measurement and we do not report kinetic measurements as determined by a method such as bio-layer interferometry (BLI), and have this adjusted the figure caption to denote the values as “apparent K_D_ values”.

Line 169 - reads strangely, "BOAS-elicited serum, regardless of its length, reactedThe length is the one of the Immunogen, not the serum

We agree that this statement is unclear, and we have modified the sentence to read:

lines 177-178:

“Each of the BOAS, regardless of its length, elicited binding titers to all matched full-length HAs representing individual components (Figure 3D).”

What is the adjuvant used (add in results)?

We used Sigma adjuvant for all immunizations, and have included this information in the results section:

lines 169-171:

“To determine immunogenicity of each BOAS, we performed a prime-boost-boost vaccination regimen in C5BL/6 mice at two-week intervals with 20µg of immunogen and adjuvanted with Sigma Adjuvant (Figure 3A).”

This information is also included in the methods section in lines 406**-**412.

Line 178 - remove " across"

We have removed the word “across” in this sentence and replaced it with “on” (line 194)

Trimer- interface, and interface epitopes are used exchangeably - maybe keep it as trimer interface to be more precise

As stated above, we have adjusted the manuscript to use the same term throughout, *i.e.*, trimer interface or its abbreviation, TI.

Line 221 - no figure 6H (6G?)

We apologize for this typo and have corrected to Figure 6G (line 231)

**Reviewer #3 (Recommendations For The Authors):**
(1) Since 20 ug x3 doses is quite a high amount of vaccine, differences between immunogens may become blurred. Thus, it may be informative to compare post-prime serology for all immunogens or select immunogens to compare to the post-3rd dose data.

We agree with the reviewer that this is on the upper end of vaccine dose and thus we explored the serum responses after a single boost. The overall trends and reactivity to matched and mis-matched BOAS components remained similar across days d28 and d42. However, the differences between the BOAS and BOAS NP groups and the mixture group were more pronounced at d28, which bolsters our claim that the presentation of the HA heads is important for eliciting strong serum responses to all components. We have included this data in Supplemental Figure 5, and have acknowledged this in the text:

lines 185-187:

“Similar binding trends were also observed with d28 serum, though the difference between the 8mer and mix groups was more pronounced at d28 (Supplemental Figure 5).”

(2) Significance statistics for all immunogenicity data should be added and discussed; it is particularly absent in Figures 3D and 7.

We have added statistical analyses to Figure 3 and Figure 7 to reflect changes in immunogenicity. We have also added the following to the methods section:

lines 482-490:

“Statistical Analysis

Significance for ELISAs and microneutralization assays were determined using either a Mann-Whitney test or a Kruskal-Wallis test with Dunn’s post-hoc test in Prism (GraphPad Prism v10.2.3) to correct for multiple comparisons. Multiple comparisons were made between each possible combination or relative to a control group, where indicated. Significance was assigned with the following: * = p<0.05, ** = p<0.01, *** = p<0.001, and **** = p<0.0001. Where conditions are compared and no significance is reported, the difference was non-significant.”

(3) Figure 2F: the figure has K03.12 listed for the H3-specific mAb and in the main text, but the caption says 3E5 - is the 3E5 in the caption a typo? 3E5 is listed for the competition ELISAs as an RBS mAb, but its binding site is distal to the RBS at residues 165-170 (https://www.ncbi.nlm.nih.gov/pmc/articles/PMC9787348/), H7.167 binds in the RBS periphery and not directly within the RBS, and the epitope for P2-D9 is undetermined/not presented. This could mean that there is actually a higher proportion of RBS-directed antibodies than what is determined from this serum competition data. Also, reference to these as 'RBS-directed' in the serum competition methods section should be revised for accuracy.

We sincerely apologize for this error and the resulting confusion. 3E5 in the caption is incorrect and should be K03.12 (https://www.rcsb.org/structure/5W08) and does engage the receptor binding site. We also apologize for the oversight that H7.167 is in the RBS periphery and not directly in the RBS. The additional P2-D9 in the panel of RBS-directed antibodies was also in error, as we do not believe it is RBS-directed, but is indeed H4 specific. We also included a reference to the paper and immunogen that elicited this antibody. We agree that this indicates that there could be a higher proportion of RBS-directed antibodies in the serum and have modified the text in the results and methods sections to read:

lines 300-306:

“Notably, this proportion is approximate, as at the time of reporting, antibodies that bind the receptor binding site of all components were not available. RBS-directed antibodies to the H4 and H9 component were not available, and the RBS-directed antibodies used targeting the other HA components have different footprints around the periphery of the RBS. Additionally, there are currently no reported influenza B TI-directed antibodies in the literature. Therefore, this may be an underestimate of the serum proportion focused to the conserved RBS and TI epitopes.”

lines 435-439:

“Following blocking with BSA in PBS-T, blocking solution was discarded and 40µL of either DPBS (no competition control), a cocktail of humanized antibodies targeting the RBS and periphery (5J8, 2G1, K03.12, H5.3, H7.167, H1209), a cocktail of humanized TI-directed antibodies (S5V2-29, D1 H1-17/H3-14, D2 H1-1/H3-1), or a negative control antibody (MEDI8852) were added at a concentration of 100µg/mL per antibody.”

(4) Only nsEM data is shown for the 3-BOAS and 8-BOAS, where differences in morphology were seen between these longer and shorter proteins. Including nsEM images for all BOAS immunogens may show trends in morphology or organization that could correlate with immune responses, e.g. if the 5-BOAS also forms a higher proportion of rosette-like structures, while the the 4-BOAS is still a mix between extended and rosette-like, this could be a factor in the better immune responses seen for 5-BOAS.

We appreciate the reviewer’s suggestion for further analysis of morphology between the intermediate BOAS sizes. We agree that the relationship between BOAS length and morphology should be explored more in depth, and we intend to do so in future studies and to also vary linker length and rigidity.